# Sensory representations in the striatum provide a temporal reference for learning and executing motor habits

Ana E. Hidalgo-Balbuena[1], Annie Y. Luma[1], Ana K. Pimentel-Farfan[1], Teresa Peña-Rangel[1] & Pavel E. Rueda-Orozco [1]

Previous studies indicate that the dorsolateral striatum (DLS) integrates sensorimotor information from cortical and thalamic regions to learn and execute motor habits. However, the exact contribution of sensory representations to this process is still unknown. Here we explore the role of the forelimb somatosensory flow in the DLS during the learning and execution of motor habits. First, we compare rhythmic somesthetic representations in the DLS and primary somatosensory cortex in anesthetized rats, and find that sequential and temporal stimuli contents are more strongly represented in the DLS. Then, using a behavioral protocol in which rats developed a stereotyped motor sequence, functional disconnection experiments, and pharmacologic and optogenetic manipulations in apprentice and expert animals, we reveal that somatosensory thalamic- and cortical-striatal pathways are indispensable for the temporal component of execution. Our results indicate that the somatosensory flow in the DLS provides the temporal reference for the development and execution of motor habits.

---

[1] Departamento de Neurobiología del Desarrollo y Neurofisiología, Instituto de Neurobiología, UNAM, Campus Juriquilla, Boulevard Juriquilla No. 3001, Querétaro 76230, Mexico. Correspondence and requests for materials should be addressed to P.E.R.-O. (email: pavel.rueda@gmail.com)

A fundamental feature of animal behavior is the ability to perform well-practiced tasks with little effort or attention. This feature, regularly referred to as automaticity or habitual execution, is generally developed after long periods of training and repetition[1]. Diverse lines of evidence indicate that the basal ganglia (BG) are instrumental in the learning and execution of motor habits[2–4], and a decline in motor automaticity is a general feature of BG-related disorders, such as Parkinson's disease[5]. It has been proposed that, as execution progressively becomes habitual, behavioral control depends on circuits involving the dorsolateral striatum (DLS)[6–9]. In rodents, the DLS receives somatotopically organized excitatory projections mainly from primary motor and sensory cortices[10,11] and from various thalamic regions[12–14]. Correspondingly, a large body of electrophysiological evidence has consistently reported somatosensory representations of different body parts in the DLS, mainly forelimbs and whiskers, during passive and active stimulation in anesthetized and awake animals[15–21]. On the other hand, it has also been reported that after learning and extensive practice, the DLS hosts "complex" behavioral representations indispensable for accurate execution, such as initiation and termination of actions[22–25], position, time, and movement speed[26–29]. Given the strong experimental evidence on the coexistence of sensory and behavioral representations, it has been proposed that this region integrates sensory information to produce successful automatic motor outputs[3]. Yet, the exact contribution of the sensory flow to the final behavioral outcome controlled by the BG is still unclear. In this context, an attractive hypothesis is that sensory representations are the scaffolding of complex representations indispensable for accurate execution, such as time or speed[30–32].

In this study, we performed anatomical mapping, electrophysiological recordings in anesthetized and freely moving animals, and permanent lesions and pharmacological and optogenetic manipulations to causally probe the role of the somatosensory flow in the DLS during the learning and execution of motor habits. We found that in rats, somesthetic representations from the forelimb arrive at the DLS not only from the canonical primary somatosensory cortex (S1)—DLS projection, but also directly from the somatosensory forelimb area of the thalamus, the ventral posterolateral nucleus (VPL). In anesthetized animals, trains of somatosensory stimulation mimicking locomotion produced neural population representations in the DLS that reflected the sequential and temporal structure of the train. In behaving animals, permanent interruption of the somatosensory flow of information to the DLS severely impaired learning by compromising the ability of the animals to extract the temporal rule in a task with spatial and temporal constraints. In expert animals performing the same task, permanent or reversible pharmacological interruptions of the sensory flow also disrupted the temporal component of execution. Finally, optogenetic manipulations of the VPL or its terminals arriving at the DLS were sufficient to bidirectionally bias the temporal component of expert execution. Importantly, the general strategy to solve the task or the basic abilities to produce stereotyped behavior or control speed were not affected. These results indicate that sensory information in the thalamo-cortical-BG loops is instrumental during both the learning and execution of motor habits by providing a temporal framework for motor commands.

## Results

### DLS receives forelimb sensory information from S1 and VPL.
Previous reports indicate that, in rodents, the DLS receives direct inputs from S1 and somatosensory regions of the thalamus, like the ventral posteromedial nucleus (VPM) and the medial posterior nucleus (POm)[14], but also from the VPL[12,13], implicated in cutaneous and proprioceptive representations of limbs and body[33]. Likewise, neural signals related to forelimb and hindlimb movements in freely moving rodents have been consistently recorded in the DLS[15,16,27,34], but the exact origin or function of these representations has not been clarified. To evaluate if in addition to the canonical S1–DLS projection, DLS also receives direct forelimb-related inputs from the thalamus, we injected the retrograde tracer Fluorogold into different regions of the DLS in six animals (Fig. 1a, Supplementary Fig. 1). Consistent with previous literature we found retrogradely labeled cells in the intralaminar (central medial and central lateral) and somatosensory regions of the thalamus including the VPM and POm but also the VPL (Fig. 1a, Supplementary Fig. 1). To confirm the anatomical results, we performed silicon probe-based electrophysiological recordings in urethane-anesthetized (1 g/kg) animals and compared response latencies in S1 and DLS to cutaneous stimulation of the forelimb contralateral to the recording sites (see Methods and Supplementary Fig. 2). We recorded 1217 cells from 12 animals; 540 cells were recorded in layers IV and V of the forelimb region of S1 (Fig. 1b; between 0.8 and 1.5 mm below the surface of the brain) and 677 cells were recorded in the DLS (between 3.5 and 4 mm below the surface of the brain) (Supplementary Fig. 2a). The following analysis was conducted in cells that significantly changed their activity between 5 and 300 ms after stimulation (216 cells in S1, 40% and 240 cells in the DLS, 35.45%; see Methods). Consistent with previous reports[35,36], cutaneous stimulation evoked complex responses in S1 (Fig. 1c left column), but also in the DLS (Fig. 1c right column). In both structures we found four types of responses: one group of cells exhibited a short-latency excitatory response; another group produced a complex pattern composed of a short-latency activation followed by a transient inactivation and a second activation; other cells responded with a transient inactivation followed by a late response between 100 and 300 ms; and finally, some cells responded with a transient inactivation. At the population level, the heterogeneity of responses to a single stimulus was reflected in a broad distribution of response latencies (Fig. 1d–f)[35–37], but importantly almost identical latencies (median 19 ms) were present in both regions (Fig. 1g, h), indicating the presence of a direct VPL–DLS input. To further explore this possibility, we expressed Channelrhodopsin-2 (ChR-II) in the VPL of rats by injecting the virus "rAAV5/CamKII-hcrR2 (H134R)-EYP" (Fig. 2a, left). One month after the infection, response latencies to light stimulation (460 nm) of VPL neuron bodies or their terminals in the DLS were recorded in the DLS (or S1) (Fig. 2a). Stimulation of the VPL produced response patterns similar to those evoked by mechanical stimulation of the contralateral forelimb (Fig. 2b), characterized by a broad distribution of response latencies (Fig. 2c), but as expected, shorter than the ones produced by mechanical stimulation (Fig. 2d, e). Here also, a subpopulation of neurons in the DLS presented short latencies overlapping with neurons recorded in S1. To directly stimulate potential fibers arriving at the DLS from the VPL, we recorded and stimulated directly on the DLS in animals expressing the ChR-II in the VPL (Fig. 2a middle). Consistent with a direct VPL- > DLS projection, we found a group of cells that robustly responded to the stimulation with short latencies (Fig. 2b, e). The interpretation of these data may be compromised by a potential spreading of the virus to other thalamic nuclei providing inputs to the DLS, such as the centrolateral complex (CL) and the parafascicular nucleus (Pf). Histological verification of the injection sites indicates that our infusions in the VPL did not contaminate the CL or Pf (Fig. 2a; Supplementary Fig. 3g, h). On the other hand, the short latencies observed when directly stimulating the VPL and the robust responses evoked by direct stimulation of the fibers in the DLS are more consistent with a direct VPL–DLS

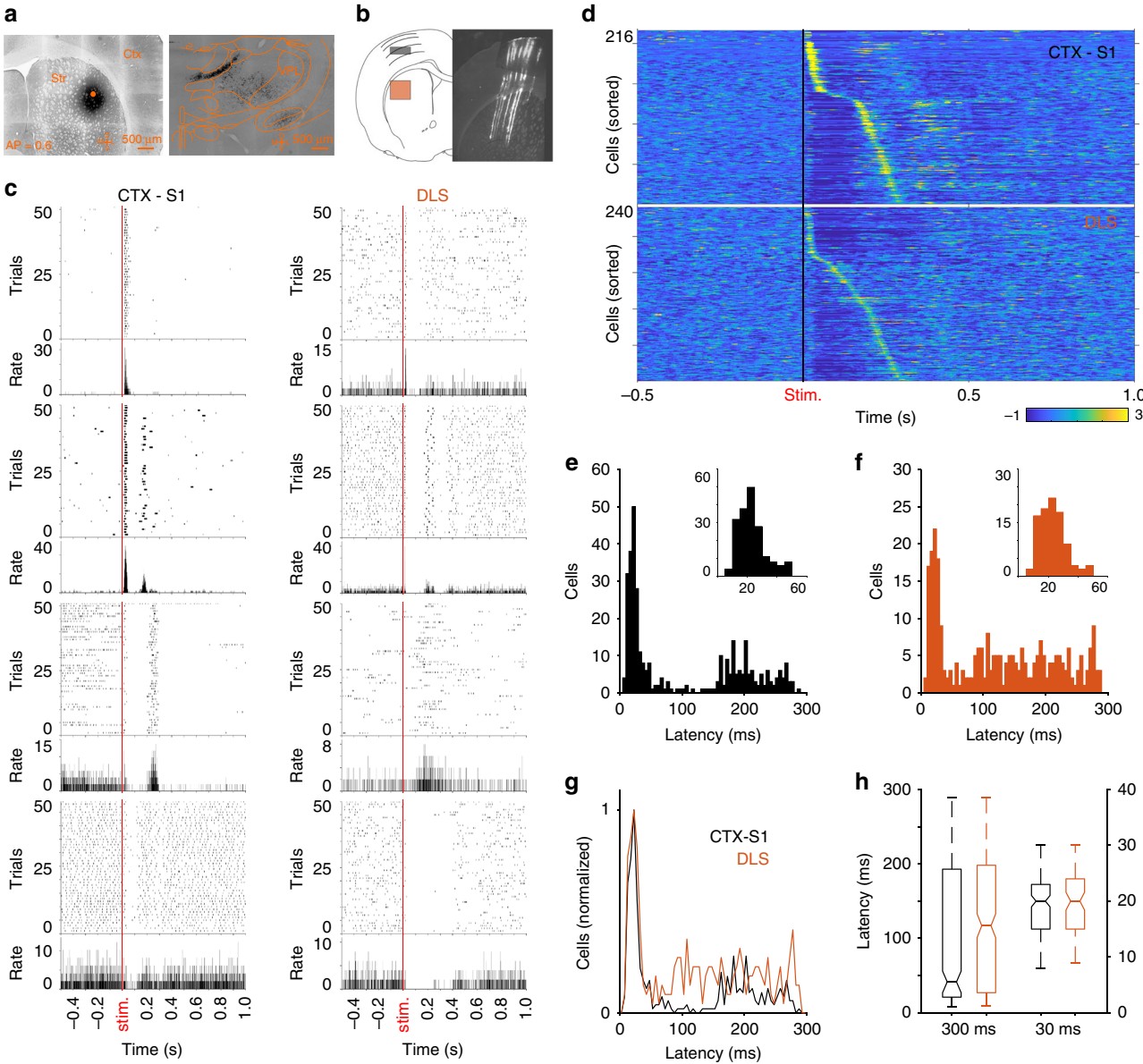

**Fig. 1** Sensory flow in the DLS and S1. **a** Micrographies show the site of injection of Fluorogold in the DLS (left), stained cells in the thalamus, including the VPL (right). **b** Schematic view of the site of recordings and histological confirmation of the silicon probe position in S1 and DLS. **c** Activity of eight illustrative units aligned to the stimulation onset of the forelimb contralateral to the recording site. Spike rasters (top) and average peri-event histograms (bottom) are depicted for cells with different response patterns recorded in S1 (left column) and DLS (right column). **d** Averaged firing rates for cells recorded in S1 (top) and DLS (bottom) expressed as Z-score (color coded) and sorted according to the time they reached the highest activity after stimulus onset. Histograms of the response latencies for all cells recorded in S1 (**e**) and DLS (**f**); for better appreciation, insets display latencies shorter than 60 ms. **g** Overlap of latency distributions in both regions. **h** Comparison of the response latencies for the entire population of cells (left) and for the cells with latencies shorter than 30 ms (right)

projection than with a multisynaptic VPL–CL/Pf–DLS projection. Finally, to discard the possibility that the short latencies observed in these experiments are related to light artifacts from local LED activation, we recorded 404 cortical and 163 striatal cells from 3 noninfected animals. These groups of cells also presented robust representations of mechanical stimulations, but no responses to direct illumination in the S1 or DLS (Supplementary Fig. 3a–f). Altogether, these results strongly indicate that the flow of forelimb somatosensory information arrives at the DLS from S1 but also directly from the VPL.

**Repetitive sensory stimulation evokes sequential activation in the DLS.** What would be the role of these forelimb-related

representations in the DLS? A previous report by Rueda-Orozco and Robbe[27] showed that in rats running on a treadmill and executing a characteristic motor sequence, spiking activity in the DLS is organized in a phasic temporal succession covering the entire execution of the motor sequence. It has also been proposed that this particular activity may constitute a temporal signal readable from the population activity[38], and perhaps a temporal framework for motor execution. To explore this possibility, first we used our anesthetized preparation. Under urethane, motor commands are significantly diminished, whereas sensory signals can be reliably evoked (Figs. 1 and 2)[20,37,39]. Hence, we could profit from a "sensory-isolated" preparation in vivo and explore the potential presence of a basic temporal structure evoked by the

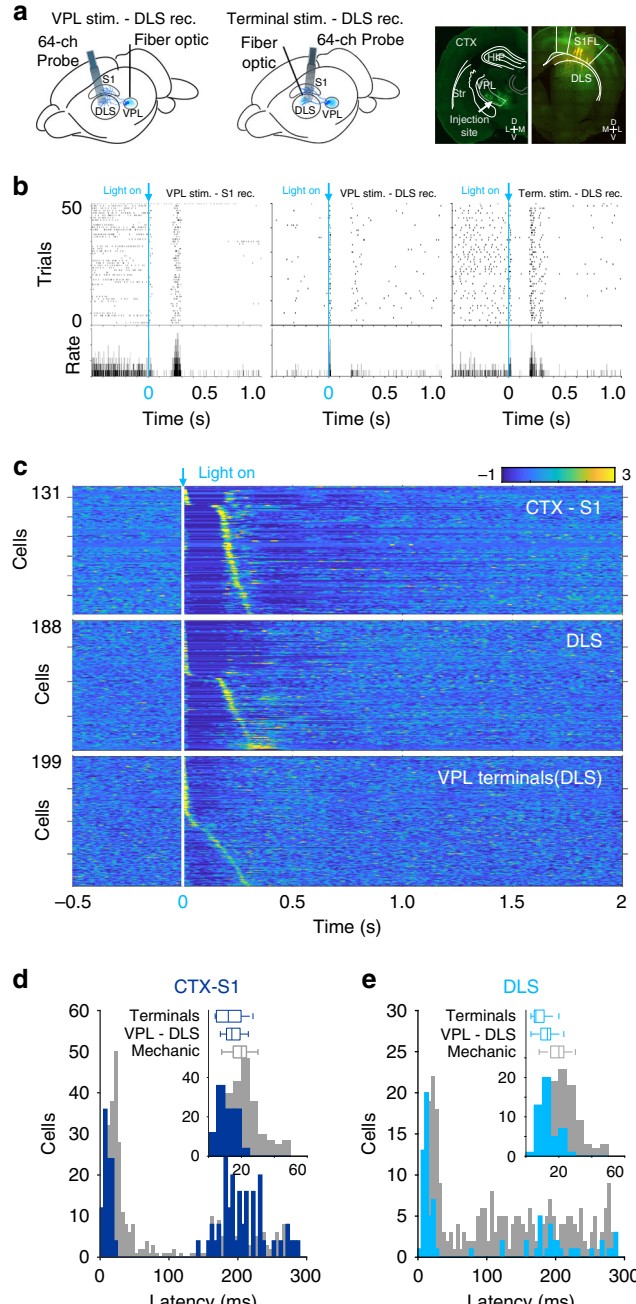

**Fig. 2** Optogenetic stimulation of somatosensory flow to the DLS.
**a** Schematic representation of experimental configuration and anatomical confirmation of infections. **b** Spike rasters (top) and average peri-event histograms (bottom) of three illustrative units aligned to the onset of optical stimulation of the VPL and recordings in S1(left) or DLS (middle), and simultaneous stimulation and recording in the DLS (right). **c** Averaged firing rates for cells recorded in S1 (top) and DLS (middle) after stimulation of the VPL or its terminals in the DLS (bottom), expressed as Z-score (color coded) and sorted according to the time they reached the highest activity after stimulus onset. Histograms of the response latencies for all cells recorded in S1 (**d**) and DLS (**e**); insets display latencies shorter than 60 ms. For comparison, the distribution of latencies evoked by somatic stimulation is presented in light gray. Box plots indicate median and 25th and 75th percentiles of the latencies evoked by somatic stimulation or stimulation of the VPL or its terminals in S1 (**d**) or DLS (**e**)

sensory flow. We used a somatosensory stimulation protocol that resembled the forelimb cycle of rats running on a treadmill at a speed of ~30 cm/s[27] (5 stimuli at 3.3 Hz; 5 ms/stimulus; 50 trains). Trains were separated by 5–10 s intervals, and stimulations were performed on the forelimb contralateral to the recording sites. We analyzed the population dynamics evoked by train stimulations in 656 cells in S1 (layers IV and V) and 579 cells in the DLS from the same 12 animals reported in the previous section. First, we aligned the spiking activity of the cells to the first stimulus of the of the train and sorted it according to the moment of the peak of their firing rate between the first and the second stimuli (Fig. 3a). To quantify the similarity in the representations evoked by each stimulus of the train, for each structure (S1 and DLS) we calculated the average population trajectory across all stimulations and then we calculated the correlation between the averaged population trajectory and the population activity evoked by each stimulus. Upon comparing the correlation values between S1 and DLS, we discovered significantly higher values in S1, indicating that the representation of each stimulus is more similar in S1 than in DLS (Supplementary Fig. 4a; Wilcoxon rank-sum test, $p < 0.001$). The robust activation evoked by each stimulus in S1 suggested that extracting a temporal or sequential component over the entire sequence of stimuli would be less informative from S1 population activity (e.g., it would be more difficult to distinguish stimulus 3 from stimulus 4 in the population activity). To explore this possibility, we applied principal component analysis (PCA) on the population activity evoked in each train of stimulation and calculated population trajectories starting at 50 ms before and finalizing 2 s after the first stimulus of the trains (Fig. 3b; Supplementary Fig. 4b, c). Then, for each train we extracted the PCA values at the time of each stimulus revealing five obvious clusters (color dots over trajectories in Fig. 3b, Supplementary Fig. 4b, c). Then we used the Silhouette method to estimate the intracluster cohesion and intercluster separations; silhouette coefficients close to 1 indicate that an element is well-classified as part of a particular cluster. For both structures, five clusters corresponding to each stimulus produced the highest silhouette coefficients, but DLS produced significantly higher scores than S1 (Fig. 3c). To validate these results, we performed the same analysis using 2–4 and 6–8 clusters; these projections produced significantly lower scores than the 5-step/cluster projection. These data indicated that each stimulus on the train produced a distinguishable sequential network state (Fig. 3b), given the possibility to decode step progression or even time. In this context, it has been previously described that striatal and cortical population dynamics are sufficient to encode information about time in in vitro and in vivo preparations[28,40–42], and these patterns of activity have been referred to as "population clocks"[30,42]. We tested if sensory stimulation in our anesthetized preparation would elicit network responses useful for decoding information about elapsed time from a given stimulus. To this aim we used a support vector machine binary decoder to classify between the time bins using the population trajectories from PCA[28] (see Methods and Supplementary Fig. 4d). We tried to decode time from the entire population of cells or from randomly selected samples increasing progressively in groups of ten cells until reaching the total number of recorded cells (Fig. 3d, see Methods). We found that prediction accuracy did not significantly improve after groups of 100 cells and that population activity in the DLS was significantly better for decoding time than S1 activity. To validate these results, we conducted the same analysis on the activity recorded during periods with no stimulation, shuffled spike trains constructed with the same number of spikes than the original data and randomly jittered spike trains. These manipulations provided prediction accuracy values that were significantly lower than the values produced with the

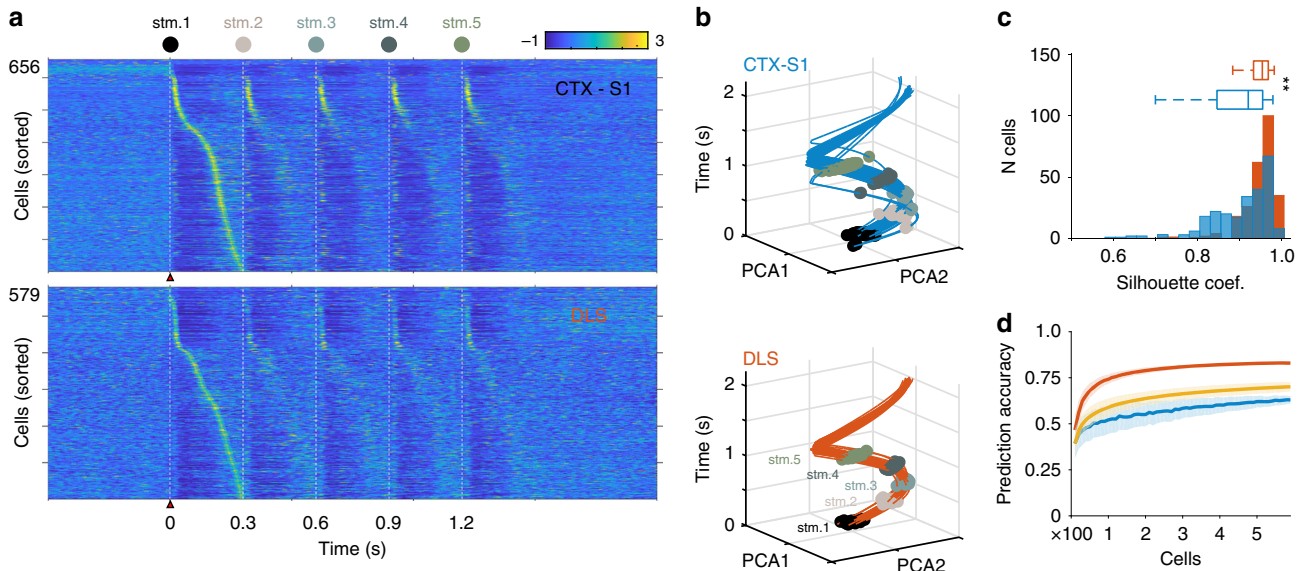

**Fig. 3** Population representations of somatosensory sequential events. **a** Averaged firing rates (Z-scores; 50 trains) for cells recorded in S1 (top) and DLS (bottom) during somatic train stimulations at 3.3 Hz. Neural activity was aligned to the first stimulus of the sequence and neurons were sorted according to the time they reached the highest firing rate between the first and second stimulus. **b** Low-dimensional projection of the population activity in S1 (top) and DLS (bottom) evoked by 50 stimulation trains. Dots correspond to the moments when the different stimuli of the train were presented. **c** Distribution of silhouette coefficients for the stimulus-evoked clusters in (**c**). ** Wilcoxon rank-sum test $p < 0.001$. **d** Comparison of Support Vector Machine classifier performance using cortical (blue), striatal (orange), and shuffled (yellow) population activity as a function of the number of units used for training and testing. Randomly selected cells were progressively added in ten-cell steps until reaching the original population size. The procedure was repeated 500 times and the mean + SD of these values is presented

population activity from the DLS (but very similar to the values obtained from S1). Finally, it is possible that the strong responses evoked immediately after each mechanical stimulation (~50 ms) could mask the decoding properties from striatal or cortical networks. To address this possible confound, we artificially removed the spiking activity in the 50 ms after each stimulus for all the cells (Supplementary Fig. 4e, f) and repeated the same analysis. The results indicate no changes in the predictive values of DLS and a slight improvement in those of S1, which did not, however, reach the levels of the DLS (Supplementary Fig. 4g). This analysis indicates that the rapid responses to mechanical stimulation did not significantly contribute to the decoding properties of these networks. Altogether, these data agree with previous observations indicating that striatal population dynamics inherently represent elapsed time[26,43,44], and suggest that a repetitive somatosensory input to the DLS would be sufficient to achieve this computation. This opens the possibility that sensory content could at least be part of a temporal framework for BG-related behaviors. We tested this possibility in a behavioral protocol where sensory representations and population sequential activation could be observed in the DLS of rats performing a stereotyped sequence of movements[27,45], as described in the following sections.

**Disruption of sensory thalamo-cortical-striatal circuits during learning.** To explore the role of the somatosensory flow to the DLS during the development of habits, we manipulated the somesthetic thalamo-cortical-striatal system in naive animals and evaluated the effects on the learning of a behavioral pattern with tight spatial and temporal constraints. We used a DLS-dependent task in which rats developed a stereotyped motor sequence with fine-tuned kinematic parameters accompanied by robust sensory and kinematic representations in the DLS[27,45]. In this task, rats running on a motorized treadmill (22 cm/s) extracted a

spatiotemporal rule to obtain a reward. When the treadmill was turned on in every trial, the animals were required to avoid (for at least 7 s, "goal time") and then enter a particular area of the environment (10 cm in the front of the treadmill, "goal area"; Fig. 4a). Correct trials (entering the goal area after the goal time) were rewarded by switching off the treadmill and giving them a drop of sucrose water. Incorrect trials (entering the goal area before the goal time) were not rewarded but rather signaled with an auditory tone (1.5 kHz) and punished with an extra 20-s run. Entrance times (moment of first entrance to the goal area) and the displacement trajectory were recorded in every trial. After long periods of training, animals developed a stereotyped "Front–Back–Front" strategy (F–B–F), the execution of which was slowly adjusted to fit the goal time (Fig. 4a). The sequence consisted in a passive displacement from the front to the rear of the running area, then a "holding period" where the animals maintained their position by running at treadmill speed and finally a last acceleration across the treadmill to enter the goal area (Fig. 4a). In this task, cyclic sensory representations from the forelimb are prominent during the running periods in the last two phases of the sequence[27]. Animals in the control group ($n = 15$) slowly developed proficient performance (>70% correct trials), characterized by the stereotyped expression of the F–B–F strategy, low variability in the entrance times and high trajectory similarity between trials (Fig. 4c–f, black lines and boxes). In a first experimental group, we aimed to permanently interrupt the forelimb-related sensory flow between the VPL and DLS by performing irreversible pharmacological lesions (see Methods) of the VPL in one hemisphere and the contralateral DLS ("VPL–cDLS", $n = 7$; Fig. 3b; Supplementary Fig. 5). This technique has been successfully used to functionally interrupt the communication between other thalamo-striatal projections[46]. We performed quantifications of brain structure with magnetic resonance imaging 10 days after the surgeries and before starting the training sessions, and discarded animals with unsatisfactory

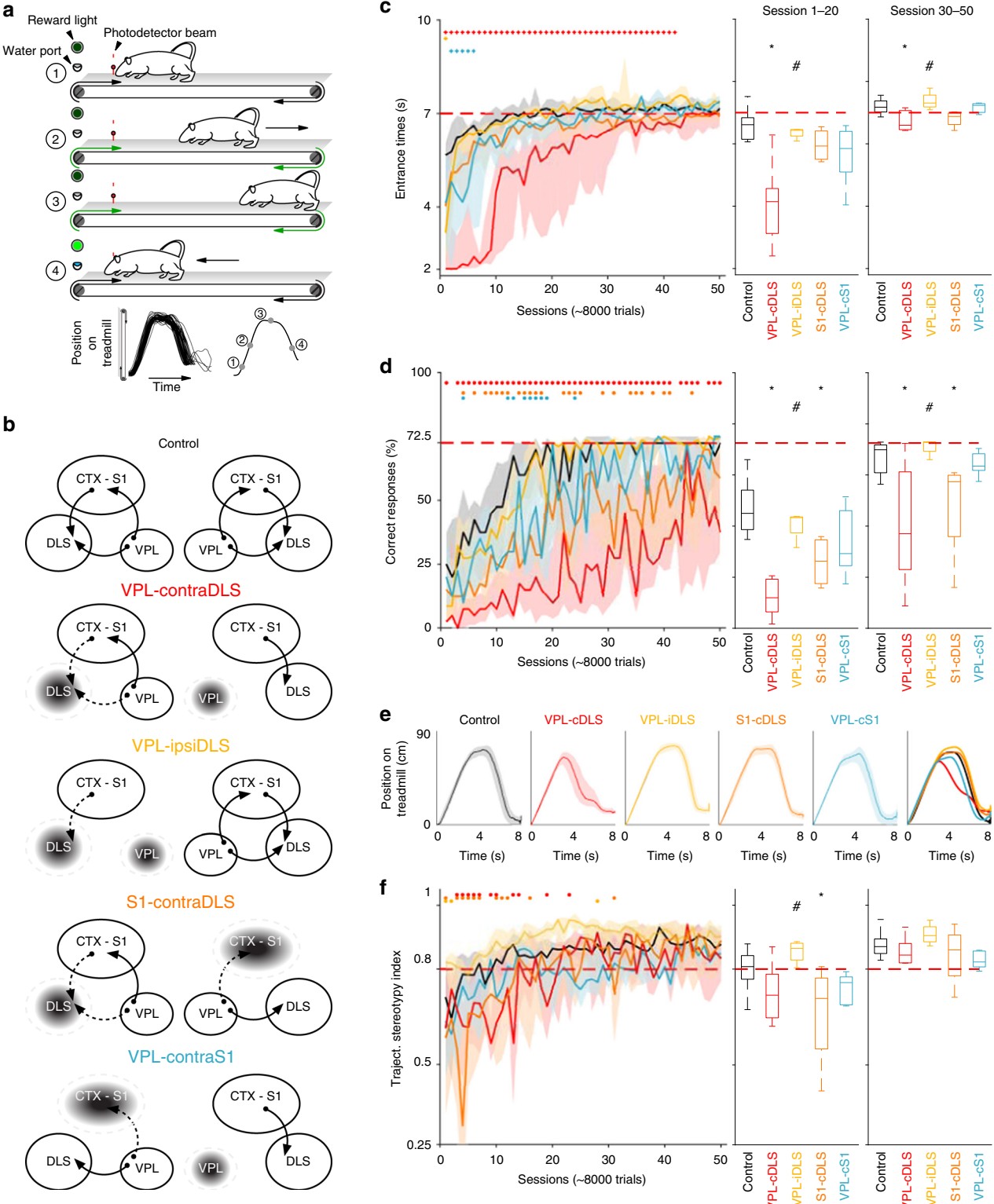

**Fig. 4** Effects of disrupting sensory flow of information to the DLS. **a** Schematic representation of the different phases of the Fron–Back–Front strategy expressed by the animals in the spatiotemporal task. Trajectories for every trial and average trajectory of a representative session are presented at the bottom. The different phases of the sequence are indicated in the average trajectory. **b** Schematic view of functional disconnections to remove forelimb somesthetic information from the DLS. Lesions are represented as dark areas in the corresponding structures. **c**, **d**, **f** Goal area entrance times (**c**), percentage of correct trials (**b**), and position stereotypy index (**f**) for the different experimental groups. Data are presented as median + 75th and 25th percentiles for the entire learning curve (left) or grouping early (middle) or late (right) sessions of training. Horizontal lines (left; two-way ANOVA, group × day of training and Bonferroni post hoc test); * and # represent significant differences vs. control and VPL–DLS groups, respectively (one-way ANOVA and Bonferroni post hoc test). **e** Representative position trajectories for one animal from each experimental group (median + 75th and 25th percentiles; color code as in (**b**))

lesions (see Methods; Supplementary Fig. 5). Electrophysiological anesthetized recordings performed in lesioned animals after completing the training confirmed that the procedure efficiently reduced forelimb sensory representations in the intact striatum (Supplementary Fig. 6). Compared with controls, animals in this group (Fig. 4 red code) presented significantly slower learning curves characterized by shorter entrance times (Fig. 4c; intragroup: CTRL $F = 11.01$, $p < 0.001$; VPL-cDLS $F = 3.75$, $p < 0.001$; intergroup CTRL vs. VPL–cDLS $F = 253.85$ $p < 0.001$) accompanied by significantly lower percentages of correct trials (Fig. 4d; intergroup CTRL vs. VPL–cDLS $F = 205.75$ $p < 0.001$). To discard the possibility that these effects resulted from particular unilateral lesions of the DLS or the VPL and not from the lack of communication between them, we lesioned both the DLS and the VPL in a different group of animals but in the same hemisphere ("VPL–ipsiDLS", $n = 4$, Fig. 4 yellow code), thus maintaining a functional DLS–VPL connection in one hemisphere. Learning in these animals was virtually identical to that of the control group, with no significant differences in percentage of correct responses in entrance times (Fig. 4c, d). Then we asked if the sensory flow to the DLS necessary for learning was arriving through the canonical VPL–S1–DLS pathway or if the VPL–DLS direct pathway was sufficient to sustain learning. To test these possibilities, in two additional groups we performed functional disconnections between S1–DLS (S1–cDLS, $n = 4$, Fig. 4 orange code) and VPL–S1 (VPL–cS1, $n = 5$, Fig. 4 blue code). Regarding entrance times, both groups presented slightly slower learning curves than the control group, but the effects were not statistically significant (Fig. 4c). The slight difference in entrance time was nevertheless sufficient to have a significantly negative impact on the percentage of correct responses in the S1–cDLS group (Fig. 4d, $p < 0.05$). Taken together these results indicate that the sensory flow of information to the DLS is necessary for the appropriate learning of the task, but the specific behavioral component that was affected remains unclear. One possibility is that the animals were unable to develop a stereotyped routine to solve the task (e.g., the F–B–F sequence) or were incapacitated to express controlled changes of speed. Another possibility is that animals were unable to precisely extract the temporal rule of the task (i.e., 7 s). To answer this question, we analyzed the trajectories of every trial per animal and calculated the correlation between every possible pair of trials in each session (trajectory stereotypy index, see Methods). With training, control animals developed the typical F–B–F stereotyped behavior characterized by a progressive decrease in variability and increase in stereotypy indexes (Fig. 4e). Surprisingly all functionally disconnected groups also developed the F–B–F strategy (Fig. 4e) and expressed similar learning curves of trajectory stereotypy (Fig. 4f; one-way repeated-measures ANOVA on day of training; CTRL $F = 4.9$; VPL–cDLS $F = 2.33$; Partial $F = 2.86$; S1–DLS $F = 2.11$; VPL–cS1 $F = 3.36$; $p < 0.01$ for all groups). The VPL–cDLS and S1–cDLS groups showed slightly lower but significant values than the control group, but these effects were circumscribed only to the early sessions of training, and during the most of the learning curve they were indistinguishable from controls (left, two-way ANOVA treatment × day of training $F = 57.54$; Bonferroni post hoc test; Kruskal–Wallis and Tukey HSD test, middle $X^2 = 12.34$, $p = 0.015$; right $X^2 = 8.7$, $p = 0.065$). Another possibility that could account for the lack of precision in the arrival times is that basic running capabilities were affected by the different lesions. To explore this possibility, we extracted the "speed trajectories" expressed by the animals during the F–B–F sequences and calculated the correlation between every possible pair of trials in each session ("speed stereotypy"). We focused our analysis on the last phase of the sequence where the animals expressed a controlled and stereotyped acceleration against the speed of the

treadmill (Fig. 5a). The speed patterns also reflected a robust intrasubject stereotypy and were not different between groups (Fig. 5a, b; Supplementary Fig. 7), indicating that the animals in all groups were capable of expressing controlled stereotyped accelerations. To assess if animals were able to properly express different speeds but without the influence of a particular stereotyped sequence of movements, we analyzed running trajectories during the punishment periods of the incorrect trials and calculated the speed occupancy patterns (Fig. 5c; Kruskal–Wallis $X^2 = 2.67$, $p = 0.44$), as well as maximum speeds (Fig. 5d; Kruskal–Wallis $X^2 = 3.74$, $p = 0.442$) and total distance covered (Fig. 5e; Kruskal–Wallis $X^2 = 8.85$, $p = 0.064$). No differences were found between groups, indicating that basic running capabilities were intact in all groups.

This set of experiments indicates that while VPL–cDLS animals experienced a slow learning curve specific to the temporal component of the behavior, their ability to produce the stereotyped F–B–F sequence or controlled changes of speed was spared. According to this interpretation, one could expect to see major effects in the "holding phase" on the back of the treadmill (Fig. 4a "3"), since this is when animals receive more rhythmic stimulation from the forelimbs. To test this possibility, we analyzed the amount of time and variance spent on each phase of the sequence (Supplementary Fig. 8). The results confirm that the most susceptible phases were the holding phase (Supplementary Fig. 8b) and the back to front phase (Supplementary Fig. 8c), in which animals in the disconnection groups significantly spent less time and presented more variability. Altogether, this set of data suggests that the sensory flow of information in the DLS is essential for providing the temporal framework for the development of the motor habit.

**Disruption of sensory thalamo-cortical-striatal circuits in expert animals.** If the sensory flow is providing a temporal reference in the DLS necessary for learning, it would be possible that these signals are also necessary for the appropriate execution of the already formed motor habits. To test this option, we overtrained rats ($n = 4$, at least 100 sessions, >10,000 trials, before manipulations) in the spatiotemporal task and evaluated the effects of reversible muscimol inactivations (50 ng/500 nl) of the VPL on the execution of the well-trained sequence (Fig. 6a). Fifteen minutes after injections, animals dramatically worsened their performance, consistently producing trials with shorter arrival times (Fig. 6b, c; $X^2 = 13.14$; $p = 0.0014$), and thus significantly decreasing the percentage of rewarded trials (Fig. 6d; $X^2 = 13.66$; $p < 0.001$). These effects were completely reversible and virtually absent in the following session (24 h after). Further analysis of the execution trajectories revealed that the animals maintained the F–B–F structure of the task, but significantly reduced the holding time at the back of the treadmill (Fig. 6e); this was directly reflected in the similar trajectory index between treated and nontreated sessions (Fig. 6f; $X^2 = 13.14$; $p = 0.023$). A closer look at the last acceleration of the animals in every trial revealed that this phase of the sequence was not significantly affected by the treatment (Fig. 6g, h; $X^2 = 4.61$; $p = 0.076$), confirming that manipulating sensory flow to the DLS did not disturb basic motor capabilities.

To further characterize the role of the sensory flow during the execution of learned motor habits, we performed permanent functional disconnections between the VPL and DLS in a different group of overtrained animals ($n = 5$; at least 100 sessions, >10,000 trials, before surgery; Supplementary Fig. 9). After lesions, subjects recovered for at least 10 days before restarting behavioral sessions. Animals in this group displayed a biphasic effect; during the first sessions after the lesion (3–4 sessions,

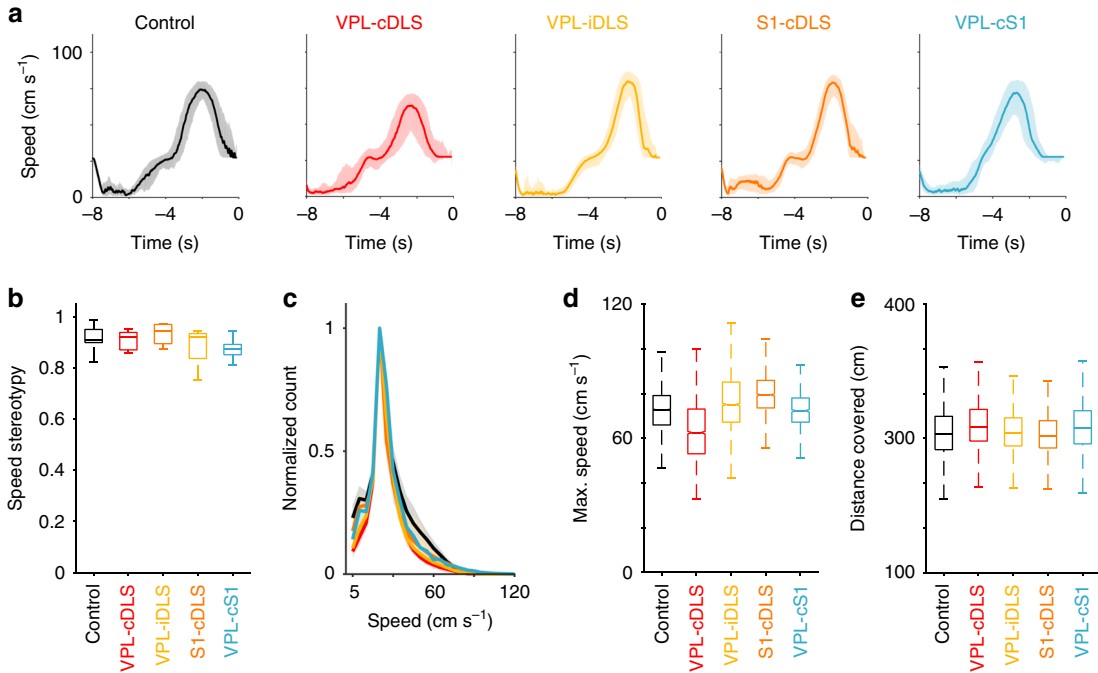

**Fig. 5** Effects of disrupting sensory flow information on speed control. **a** Speed trajectories (median + 75th and 25th percentiles) for one representative animal from each experimental group (color code as in Fig. 4). **b** Speed stereotypy index for all experimental groups (median + 75th and 25th percentiles). **c** Normalized distribution of running speeds (median + 75th and 25th percentiles) expressed during the punishment periods of incorrect trials for the different experimental groups. Maximum speeds (**d**) and distance traveled (**e**) for the same conditions as in (**c**) (median + 75th and 25th percentiles)

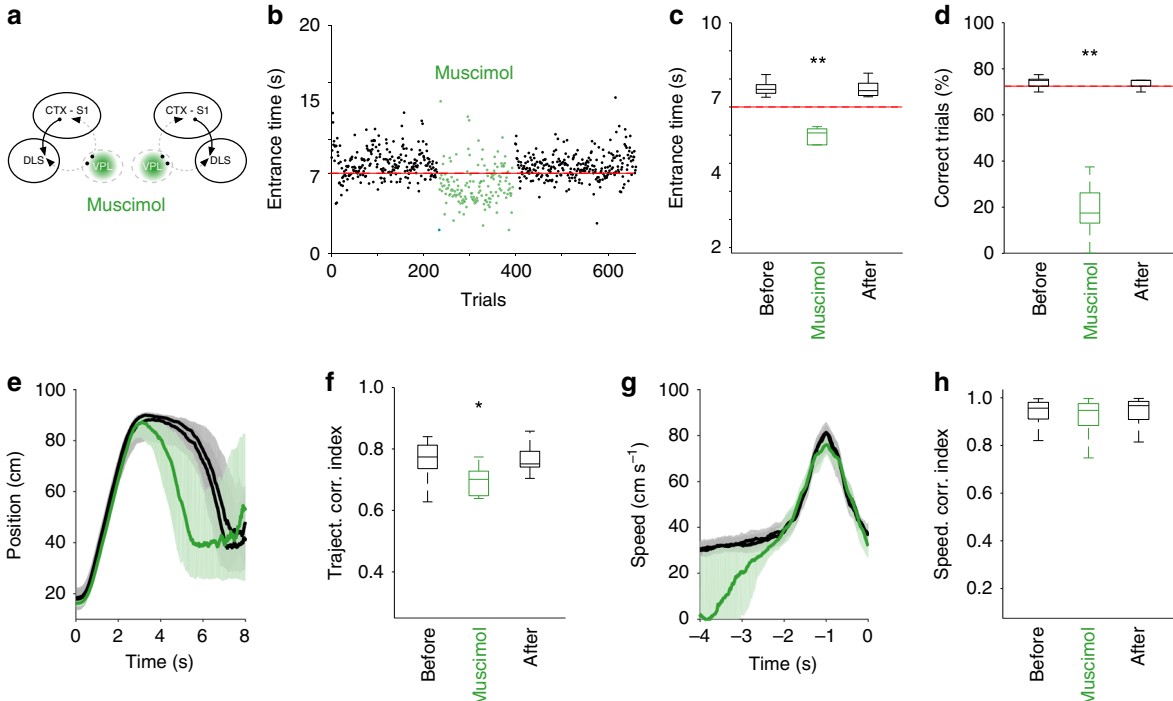

**Fig. 6** Effects of pharmacological inactivation of theVPL. **a** Schematic representation of the VPL inactivation procedure. **b** Goal area entrance times for four sessions before and after (black) the administration of muscimol in the VPL (green, 50 ng in 0.5 μl). The different phases of the sequence are indicated in the average trajectory. **c** Group effect for muscimol injections on entrance times (**c**) and percentage of correct trials (**d**) (median + 75th and 25th percentiles). Representative position (**e**) and speed (**g**) trajectories and group effects on position (**f**) and speed (**h**) before and after (black) muscimol (green). *$P < 0.01$, **$P < 0.001$, Kruskall–Wallis and Tukey's HSD test (**c**, **d**, **f**, **h**)

around 500 trials), all animals experienced a decrease in performance similar to the one observed after transient inactivation of the VPL, that is, significantly shorter arrival times accompanied by a decrease in the percentage of correct trials (Supplementary Fig. 9a, b). Interestingly, after this period all animals recovered the ability to arrive after the goal time but with significantly higher levels of variability (Supplementary Fig. 9c). Animal's performance was followed for at least 30 more sessions during which the level of variability never recovered the prelesion values (Supplementary Fig. 9a–c). Analysis of the execution trajectories confirmed that animals did not lose the F–B–F strategy (Supplementary Fig. 9e, g); and they maintained virtually identical dynamics in the last acceleration of the sequences, indicating that the increased variability in arrival times was not the result of impaired motor capabilities (Supplementary Fig. 9g–i). This set of data confirmed that the flow of information to the DLS is instrumental for the timed execution of already formed motor habits.

**Optogenetic manipulation of the sensory flow to the DLS.** Altogether, our data suggest that somatosensory information in the DLS may act as a temporal reference for the appropriate development of timed execution of a motor habit. Therefore, we hypothesize that actively modifying the sensory flow to the DLS would be enough to bias the temporal component of execution. To test this hypothesis, in another group of overtrained animals (>100 training sessions and >10,000 trials, before infections), we performed optogenetic manipulations of sensory inputs to the DLS. First, to corroborate the previous results, we infected the VPL bilaterally with the virus rAA5/CamKIIa-eNPHR3.0 to express Halorhodopsin or the virus rAA5/CamKIIa-eArchT3.0 to express Archaerhodopsin and implanted optical fibers over the site of infection in the VPL (two animals) or its terminals in the DLS (two animals) (Fig. 7a). To test these animals (and all the animals from the following experiments), during multiple sessions we illuminated the VPL (or its terminals in the DLS; 550 nm, 9.2 mW for Halorhodopsin or 525 nm, 6.2 mW for Archaerhodopsin) in 50% of randomly selected trials. We performed continuous stimulation specifically during the holding and last acceleration phases of the sequence (Fig. 7a, bottom). We chose these phases since they are the moments where sensory representations from the cyclic movement of forelimbs are more prominent during the execution of the task[27]. A comparison of the stimulated vs. nonstimulated trials showed similar effects to those of muscimol experiments; that is, during stimulated trials animals presented shorter arrival times that consequently resulted in significantly lower percentages of correct trials (Fig. 7b, c). Like in the previous cases, the animals maintained the F–B–F strategy and no effects were observed in the last acceleration of the sequences (Fig. 7d). Importantly, stimulation in the VPL terminals in the DLS produced the same effects as the stimulation of the VPL itself, confirming that manipulating the VPL–DLS direct projection is sufficient to bias the timed execution of the well-learned motor sequence. Up until this point, every manipulation we performed aimed at permanently or transiently removing the sensory flow and resulted in a very similar behavioral outcome: a decrease in the arrival times. This suggests that if we could facilitate or increase the frequency of the sensory flow, then we would obtain the opposite result: an increase in arrival times. To test this possibility, we expressed ChR-II in the VPL of five animals and we implanted fibers over the site of infection (Fig. 7e). Like in the previous case, we illuminated the VPL (465 nm; 0.5–2 mW), but this time we used continuous and train stimulations at 5, 3.3, 2, and 1 Hz (Fig. 7e bottom; see Methods). Train protocols produced a consistent effect on the five animals: an increase in

arrival times and hence an increase in correct responses (Fig. 7f, g and Supplementary Fig. 10f for 3.3 Hz; other frequencies in Supplementary Fig. 10a; K–W Rat A $X^2 = 231.12$, $p < 0.001$; Rat B $X^2 = 241.12$, $p < 0.001$; Rat C $X^2 = 343.12$, $p < 0.001$). None of the protocols changed the F–B–F strategy or produced significant changes in the last acceleration of the sequences (Fig. 7h). Continuous stimulation produced significant effects in four of the five animals, and these effects were similar to the ones evoked by train stimulation (Supplementary Fig. 10a). These results confirmed that bidirectional manipulations of the sensory flow had an impact on timed execution. On the other hand, our anatomical, electrophysiological (Figs. 1 and 2), and lesion experiments (Fig. 4) suggested that manipulating the VPL–DLS projection would be enough to affect the temporal component of the execution. To specifically stimulate the VPL–DLS pathway, we expressed ChR-II in the VPL and implanted optical fibers in the DLS (Fig. 7e) of three overtrained animals. Under these conditions, and using the stimulation protocol described above, we found that train stimulations produced the same behavioral pattern as the one observed with stimulations directly on the VPL; that is, major effects at 3.3 Hz (Fig. 7f, g; K–W Rat D $X^2 = 337.68$, $p < 0.001$; Rat E $X^2 = 350.75$, $p < 0.001$) and milder but significant effects at 5 and 2 Hz, with no effects at 1 Hz or continuous stimulations (Supplementary Fig. 10a). Similarly, the F–B–F strategy and the kinematics of the last acceleration of the sequences were not significantly changed by either stimulation. An important observation is that, lesions, pharmacological, or optogenetic manipulations had little impact in the last phase of the sequence of movements, in which animals expressed a stereotypical acceleration across the treadmill. These results indicate that the main effects of our treatments occurred in the holding phase of the sequence (Supplementary Fig. 8). To confirm this possibility, in two animals we specifically stimulated the last phase of the sequence (Supplementary Fig. 10b–d). In these experiments, stimulations did not produce significant changes in arrival times (Supplementary Fig. 10c) or percentage of correct responses (Supplementary Fig. 7d). In the same line, a comparison of the times spent in each phase of the sequences between stimulated and nonstimulated trials for both inhibition and excitation animals (Supplementary Fig. 11) indicated that the main effects of the stimulations were specific to the holding phase of the sequence. Further confirming the specificity of our optic manipulations, one of the animals used in this section (Rat H; Supplementary Fig. 10e–h) was incorrectly infected; histology revealed that the virus was deposited in the left hemisphere, between the ventral region of the laterodorsal thalamic nucleus and the dorsal part of the posterior thalamic group, not the VPL (Supplementary Fig. 10e). This animal presented no behavioral effects in any of the stimulation protocols (Supplementary Fig. 10f–h).

Finally, to determine if the optical manipulations functionally affected the forelimb representations during the execution of the task, we performed freely moving silicon probe recordings during sequence execution in three expert animals and one naïve animal subjected to irreversible pharmacological lesions in the VPL and the contralateral DLS. Expert animals were infused in the VPL with the virus to express ChR-II and also implanted with two optical fibers directed to the VPL. In expert animals we recorded 176 cells from the forelimb region of S1 (131 cells from the lesioned animal) and 94 cells from the DLS. During recording sessions, 50% of randomly selected trials were optically stimulated, producing behavioral effects very similar to those in the animals without silicon probes (Supplementary Fig. 10b–d). In expert animals, firing rates changed significantly between stimulated and not-stimulated trials (Supplementary Fig. 12a–c), the auto-correlograms of many cells presented rhythmic

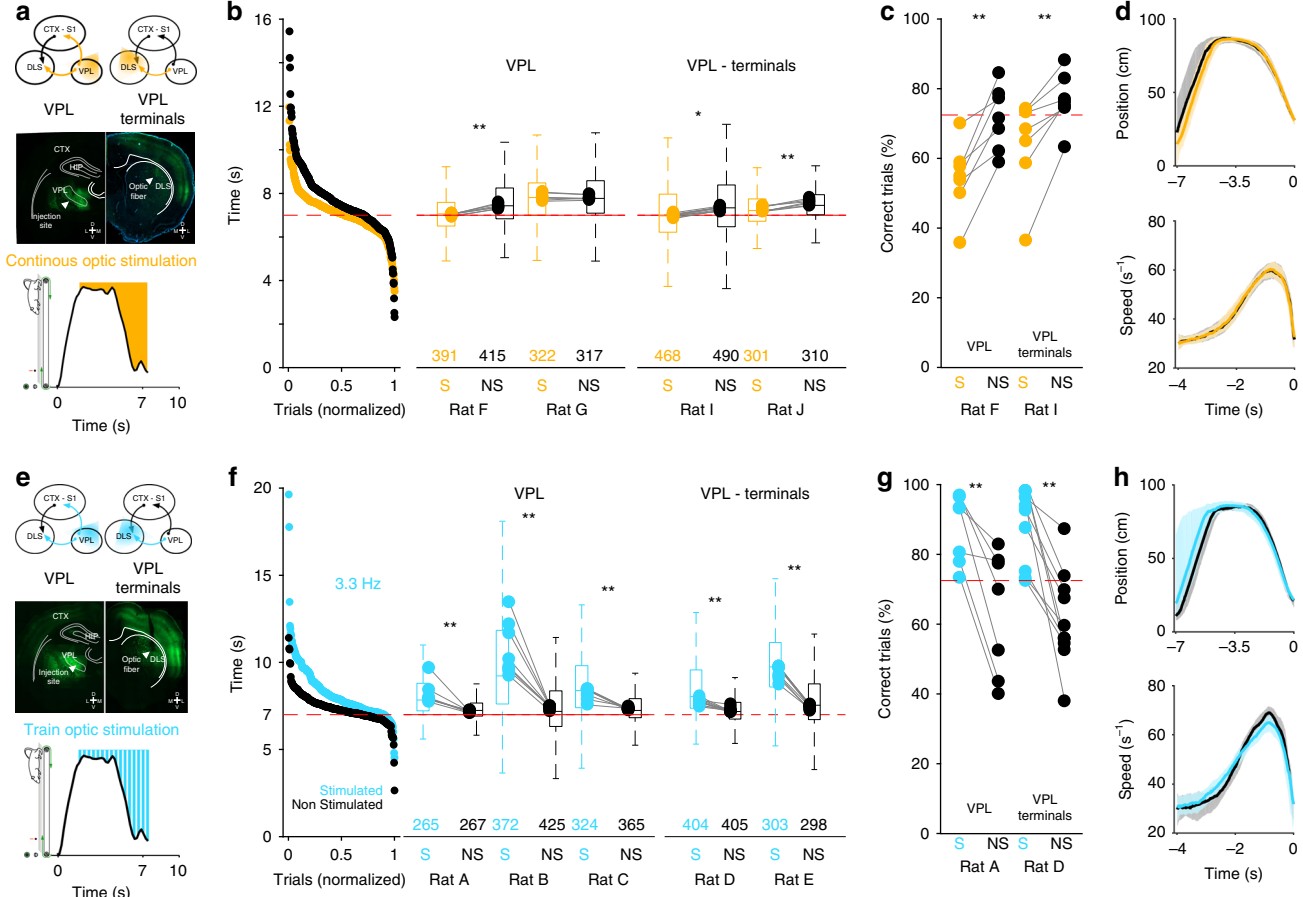

**Fig. 7** Effects of optogenetic manipulation of VPL or its terminals. Schematic representation of the infection and stimulation sites (**a**, **e** top). Histological confirmation of viral expression and optical fiber positions in the VPL and DLS (**a**, **e**, middle). Schematic representation of the moment when the stimulation was delivered during 50% of the randomly selected trials (**a**, **e**, bottom). **b**, **f** Entrance times for all stimulated (yellow for Halorhodopsin or Archaerhodopsin; blue for Channelrhodopsin-2) vs. nonstimulated (black) trials; box plots indicate the median (25th and 75th percentiles) of the stimulated vs. nonstimulated trials; pairs of dots united by lines indicate medians of each session for each animal; the number of trials analyzed for each animal and condition is indicated at the bottom. **c**, **g** Percentage of correct trials for one animal on each group. **d**, **h** Average position (top) and speed (bottom) trajectories for all stimulated and non-stimulated trials of Rat F (**d**) and Rat A (**h**). **Wilcoxon rank-sum test $p < 0.001$

modulation around 3.3 Hz (corresponding to the forelimb cycle when rats run at 30 cm/s; Supplementary Fig. 12d) and the correlation between the spiking activity of individual cells and the movement of the forelimb showed correlation distributions with a peak toward high-correlation values above 0.6 for both S1 and DLS (Supplementary Fig. 12e, f). The animal with the lesion in the VPL showed a significantly different distribution with the majority of the cells expressing lower correlation values (Supplementary Fig. 12e, g). Then we compared only the cells with the higher correlation values during nonstimulated trials and found that optical stimulation of the VPL significantly decreased the correlation values in both S1 and DLS (Supplementary Fig. 12e, g). These results indicate that optical stimulation of the VPL efficiently changed the sensory representations of the forelimb in both, S1 and DLS.

## Discussion

The presence of somatosensory representations in the DLS has been previously reported in different behavioral circumstances[15–19,34], including the execution of motor habits[27], but the exact contribution of these representations to the final behavioral outcome remained elusive. Here, we have explicitly explored the role of the somatosensory flow in the DLS during the learning and execution

of motor habits. By combining anatomical quantifications, electrophysiological recordings in anesthetized and freely moving animals, as well as lesion experiments and pharmacological and optogenetic manipulations, we have provided evidence indicating that somatosensory information in the DLS is essential for the learning and execution of motor habits by providing a temporal framework for motor commands.

Previous reports indicate that aside from the canonical cortical-striatal projection, sensory representations from different modalities may also arrive at the BG directly from sensory-related thalamic nuclei[12–14] and from the CL/Pf complex[47,48], the main source of thalamo-striatal information[49]. While we cannot rule out the possibility that forelimb-related information in the DLS is also arriving from the CL/Pf complex, our experiments confirms a major contribution of the direct VPL–DLS projection. First, mechanical stimulation of the forelimb produced overlapping short-latency responses in S1 and DLS (Fig. 1). Second, short-latency responses to light stimulation in the DLS in animals expressing ChR-II in the VPL (Fig. 2) confirmed the presence of functional terminals and are incompatible with the possibility of backpropagated activation of the VPL, which would in turn, activate S1 or CL/Pf and finally the DLS.

What would be the role of these representations? Previous reports in primates have demonstrated that aside from the

sensory representations in the putamen, individual cells encode relevant information related to categorical decisions or the initiation of specific actions based on specific patterns of somatosensory stimulation[50,51]. Of particular relevance, Liles[52] has demonstrated that, also in the putamen, individual cells may integrate both sensory and motor signals, opening the possibility of a direct influence of sensory information over motor commands. Despite the importance of these observations, authors did not directly explore the relevance of sensory signals in the construction of complex representations, such as time. Nor did they directly link the sensory information with specific motor commands or sequences of movements. On the other hand, experimental and theoretical evidence support the notion that timing may arise from reproducible time-varying patterns of activity evoked by the interaction between neural networks, such as the cortex or the striatum, and incoming events such as sensory inputs[30,40]. In our experiments the use of an anesthetized preparation enabled us to record sensory pathways and diminish potential motor-related confounds. Under these conditions, trains of mechanical stimulation in the forelimb produced robust time-evolving population patterns of activity in S1 and DLS (Fig. 3). However, activity in the DLS was significantly more useful for decoding sequential events (such as each stimulus on the train) and elapsed time. These data are consistent with previous observations indicating that population dynamics in the striatum reflect the animal's sensorimotor state and are necessary for interval estimations[26,28,43]. An important question is whether under these anesthetized conditions, sensory inputs to the DLS may sustain temporal representations in longer timescales, like those reported for the range of dozens of seconds to minutes[26,43]. This possibility would need to be addressed experimentally but it is likely that those longer temporal representations are based on a more complex mechanism, since they were reported in conditions were animals were not necessarily implicated in a particular sequence of movements or a stereotyped behavior that would produce a stable, rhythmic sensory flow. In this regard, our data are more compatible with the idea that sensory-based temporal representations are fundamental for the learning and execution of short sequences of movement in the range of seconds. Moreover, our data support the notion of a basic population organization that may be recruited—in this case by somatosensory events—for the representation of relevant behavioral parameters, such as time[37,38,53,54].

To explore the function of these representations in motor habits, we used an ad hoc behavioral protocol where animals developed a stereotyped sequence of movements characterized by a sequence of changes of speed constrained by a temporal and a spatial rule. Our behavioral observations also tallied with a temporal-related function for the sensory representations in the DLS. Animals subjected to VPL–cDLS lesions (Fig. 4b) were severely impaired in extracting the temporal rule of the task, resulting in slow learning curves specifically in the temporal domain (Fig. 4). Lesions of S1–DLS and VPL–S1 pathways resulted in milder behavioral deficits, indicating that the direct VPL–DLS projection would be enough to provide a temporal context while running on the treadmill. Similar behavioral effects were obtained in expert animals by pharmacological inactivations (Fig. 6) or permanent lesions (Supplementary Fig. 6), suggesting that this timing-related function remains stable through the course of learning. A fundamental observation in this set of data is that, while the temporal component of behavior was affected, a basic analysis of the quality of the behavior and speed indicated that the ability to generate the stereotyped F–B–F strategy (Fig. 4e, d) or to control speed (Fig. 5) was spared in all groups. More importantly, the last acceleration of the sequence of movements (Fig. 4a), which required greater motor control, was

virtually identical in all conditions (Fig. 5), confirming that the deficits induced by these manipulations were specific to the temporal component of the behavior. Optogenetic experiments in expert animals strengthened this notion by confirming two things: that the temporal component of behavior could be selectively and bidirectionally biased by manipulating the sensory flow, and that the activation of the VPL–DLS projection was sufficient to achieve these effects. The different manipulations also indicate that the holding phase of the sequence is when sensory information is more relevant. This idea is supported by the fact that specific optogenetic stimulations in the last acceleration phase did not interfered with performance (Supplementary Fig. 10b–d). Some important issues are still to be considered in our interpretations; for example, pathway specificity during the learning curve. Despite that S1–cDLS and VPL–cS1 lesions produced significantly milder effects than VPL–cDLS lesions (Fig. 4), we are not in a position to claim that we achieved a perfect VPL–DLS pathway-specific disconnection; for instance, in this group both ipsilateral S1 and DLS would result without sensory inputs (Supplementary Fig. 6). Another example is that optogenetic stimulation at different frequencies did not linearly increase the arrival times. This could be explained by the fact that our manipulations did not substitute the sensory feedback produced by the natural movement of the forelimbs. Further experiments to exactly understand the effects of these kinds of manipulations on circuit dynamics will be conducted in future investigations.

Our interpretations are in line with previous experimental data in primates[55], which were trained to produce forelimb motor commands, synchronized to an auditory "metronome" for three cycles with a fixed interval (e.g., 450 ms). After the three cycles of synchronization, the animals were required to produce three more cycles with the same interval but in the absence of the auditory metronome. The authors report that as the sensory cues disappear, animals tend to produce faster intervals. The shortening of the interval was more noticeable when animals were required to produce longer intervals, e.g., 1000 ms. Interestingly, in this time scale animals are required to produce 6 intervals (3 synchronizations + 3 continuations = ~6 s), which is similar to the sequence execution in the present task for rodents (7 s). Our observations are also in line with previous reports in rodents, in that the sequential activation of cells evoked during the execution of the treadmill task remained stable at early and late stages of learning[27]; this type of activity has been proposed as a potential mechanism to represent time in recurrent networks such as the striatum[26,38,43] and cortex[28,40]. Although the exact origins of sequential activation and their relationship with individual cells encoding different parameters (e.g., space and position) remain to be determined, here we have shown that sensory stimulation can be a significant contributor. In this paper we have exclusively studied the participation of forelimb somatosensory representations because of their prominent presence in the DLS during locomotion in rodents[27,34,56] (Supplementary Fig. 12). Nevertheless, previous literature has provided strong functional[52,57] and anatomical[58,59] evidence of the convergence of sensory and motor inputs onto specific striatal cells. Moreover, it has been previously proposed that a basic temporal structure represented in population sequential events could emerge from different motor and sensory cortical and thalamic inputs to the BG[26,38]; thus, to fully understand the integrative functions of the DLS, it would be necessary to clarify the interactions with other sensory representations, such as the whisker[14] or visual[20] systems, and more importantly with motor-related inputs. Finally, the data presented herein further support previous proposals[32,60] indicating that one of the main

functions of the striatum is to continuously monitor the sensorimotor state of the animal to contextualize the following motor commands.

## Methods

**Ethical approval.** All experimental procedures were approved by the Animal Ethics Committee of the Institute of Neurobiology, National Autonomous University of Mexico (UNAM) and conformed to the principles outlined in the Guide for the Care and Use of Laboratory Animals (NIH). All efforts were made to minimize the number of animals used and their suffering.

**Animals.** Long-Evans rats ($n = 54$; 250–700 g) where housed in pairs at a stable temperature (23 °C) and humidity (66%) under a constant 12:12-h light-dark cycle (lights on at 8 a.m.) and with free access to food and water. All experimental procedures were conducted during the light phase of the cycle. Six animals were used for path tracing experiments with Fluorogold (FG). Twelve animals were used for anesthetized electrophysiological experiments. Thirty-six animals were trained in the spatiotemporal task and used for lesion, pharmacological, optogenetic, or freely moving recordings. Animals were bred and maintained in the satellite bioterium of our laboratory and constantly supervised by specialized personnel from the general animal facility of our Institute.

**Retrograde tracing.** Tracer, viral and NMDA injections were performed using Hamilton Neuros syringes (1 μl). A total of 100–300 nl of Fluorogold (5%) was dissolved in 0.9% NaCl. Injections were performed in five different regions of the DLS (Supplementary Fig. 1) at a rate of 0.2 μl/min. After 10 days of survival, animals were transcardially perfused with PBS followed by 4% paraformaldehyde. Coronal slides (50 μm) containing the striatum and thalamus were obtained by means of a vibratome (Leica) and visualized and photographed under fluorescence microscopy. Semiautomatic quantifications of the injection volumes in the DLS and number of cells in the VPL were obtained with custom-made routines in Matlab and Labview.

**Anesthetized experiments.** Animals were anesthetized with urethane (1 g/kg) and mounted on the stereotaxic frame; supplemental doses (0.15 g/kg) were given when necessary after hours of the initial dose. Silicon probe recordings were performed through a craniotomy (2 mm × 2 mm) centered at 0.6 mm anterior and 3.7 mm lateral to bregma (Supplementary Fig. 2a). The stereotaxic coordinates for recordings were selected based on previous literature indicating that these regions in S1 and DLS contain principal representations of the forelimb[15,35,61]. Moreover, before starting any experiment, and upon placing the recording probes in the brain, we inspected the local field potential response evoked by the forelimb and hindlimb stimulations, confirming that all our recordings were performed in the forelimb region (Supplementary Fig. 2b, c). Histological confirmation of silicon probe positions in the brain was achieved by applying DiI lipophilic carbocyanine dye (1%; Sigma) to the back of the probes (Fig. 2a). For optogenetic experiments an optical fiber (diameter 200 μm) was fixed on the VPL 100 microns over the infection site or directly on the DLS or S1 ~100 μm away from the recording sites. Probes were slowly inserted into deep layers of the cortex (depth 900–1500 μm) and the striatum (depth 3000–4500 μm). For each animal we performed recordings in 2–3 depths of the cortex and 3–5 depths in the striatum separated by at least 250 μm. Somesthetic stimulations were performed on the pads of the contralateral forelimb to the recording site with a solenoid valve attached to a cotton tip. The cotton tip was fixed to stimulate the palm of the paw, where it touched at least four pads (Supplementary Fig. 3b). Stimulations aimed to mimic the experience of the animals when they walk or run; hence, we did not intend to separate somatic from proprioceptive stimulation. Nevertheless, stimulation did not produce evident movements of the entire limb. Each stimulus was presented for 5 ms, and stimulation was given every 5 s as single stimulus or as trains of 5 stimuli (3.3 Hz). In some animals, mechanical stimulation of the pads was compared to electric stimulation of the forepaw pads (electrodes in medial and lateral pads). Both stimulations produced very similar responses (Supplementary Fig. 2b, c). Immediately after the experiments, animals were injected with a lethal dose of pentobarbital and transcardially perfused; their brains were then extracted and processed for histological quantifications.

**Implantation of silicon probes for unit recordings in behaving animals.** Four animals were used for this type of recordings. One month (three animals) or 10 days (one animal) before this procedure, animals were subjected to a surgery in which the virus to express ChR-II (three animals) or NMDA (one animal) was infused in the VPL (and the contralateral DLS for the NMDA-treated animal). Stereotaxic coordinates for these injections are described in the following sections. Under deep sevoflurane anesthesia, 64-channel silicon probes (Neuronexus, Buzsaki-64 or Tet 4 × 4) were implanted in S1 above the striatum (probe centered at: AP = 0.6, DL = 3.7 mm with respect to bregma; Supplementary Fig. 2a). Two miniature screws implanted above the cerebellum served as ground and reference. After animals recovered from surgery (at least 10 days) the probe was lowered toward the recording sites (50–200 μm d$^{-1}$). In the same surgery, two optical fibers directed to the VPL of

the three infected animals were implanted in the following coordinates in millimeters with respect to bregma: AP = − 2.3; DL = ± 2.8; V = − 6.1 (500 μm above the site of virus infection). For well-trained animals recordings in S1 were performed in 9 different depths between 0.9 and 1.9 mm, and in the DLS in 12 different depths between 3.1 and 4.1 mm during 12 sessions. For the naïve animal we recorded 131 cells from 12 different depths between 0.9 and 2.2 mm during 6 sessions.

**Electrophysiological data acquisition and processing.** Wide band (0.1–8000 Hz) neurophysiological signals from silicon probes (Neuronexus, Buzsaki-64) were amplified 1000 times via Intan RHD2000-series Amplifier evaluation system or Plexon VLSI headstages and a PBX2 amplifier and continuously acquired at 20 kHz on two synchronized National Instruments A/D cards (PXie 6363, 16-bit resolution). Spike sorting was performed semi-automatically using the clustering software KlustaKwik (http://klustakwik.sourceforge.net) and the graphical spike sorting application Klusters (http://klusters.sourceforge.net)[62].

**Analysis of the neural data.** To determine the response latencies to mechanical or optical stimulation, the neuronal data was binarized to a resolution of 1 ms. Peri-event histograms were constructed and a confidence limit of 99% based on baseline activity (−1 s before each stimulation) was calculated for each cell. Responses to stimulation were considered significant if they exceeded the confidence limit by at least 1 ms, and the time of the first bin was defined as the response latency.

For low-dimensional representation of population activity, we *first* created temporal population vectors (50-ms nonoverlapping bins) containing the firing rates of each neuron (dimension) of the population within a 3-s window from −1 to 2 s from the first stimulus of the train. This binarization rendered a total of 60 population vectors, one for each 50-ms bin. Then we used standard PCA to reduce dimensions and plotted the 2 first principal components reflecting the progression of the population state as function of time (Fig. 3c), or the different combinations between principal components 1 and 5 (Supplementary Fig. 4b, c). Next, we extracted the population projections at the times when every stimulus of the train was presented, resulting in 250 population states ("observed projection", 5 stimulus × 50 trains). To evaluate if population states formed segregated clusters corresponding to each stimulus of the train, we randomly assigned each population state to projections with 3–7 clusters, calculated the Silhouette Coefficient (SC) for each state and compared with SCs obtained from the observed projection. $SC = (d' − d)/\max(d, d')$, where $d$ represents the mean intracluster Euclidean distance and $d'$ the mean nearest-cluster Euclidean distance to the considered state. Values close to 1 mean perfect assignment to its cluster; negative values indicate that the state was wrongly assigned to the cluster and values close to zero indicate that the state is equally likely to belong to its cluster or the nearest one. This analysis was performed independently for S1 and DLS data.

For the population decoder, we decoded elapsed time from the PCs extracted from the cortical or striatal population activity using a Support Vector Machine algorithm. We used the one-against-one approach as in ref. [28]. First, 20 population trajectories were randomly assigned as "training trajectories" and the remaining 30 were assigned as "comparison trajectories" (Supplementary Fig. 4d panel i). Then each population trajectory was binarized into 50-ms population time bins (Supplementary Fig. 4d panel ii). The time bins from the "training trajectories" were used to train a binary classifier to distinguish between every possible pair of population time-bins (Supplementary Fig. 4d panel iii) rendering 60 informative bins (Supplementary Fig. 4d panel iv). After the training, the classifier labeled each of population time-bins from the 30 "comparison trajectories", as belonging to one of the sixty 50-ms informative bins, rendering 60 classifying scores, one for each time bin (Supplementary Fig. 4d panel v). Finally, the 60 scores were averaged. To determine how the size of the neural population affected the ability of the network to classify time bins, we replicated the analysis in randomly selected pools of cells in increasing steps of 10 cells until reaching the total number of cells in the population (656 in S1 and 579 in DLS). These procedures were repeated 500 times, thus producing the medians and 25th and 75th percentiles reported in Fig. 3e for S1 (blue) and DLS (orange). Then we created confidence intervals by running the same analysis in 3-s periods where no stimulation was provided; these periods were randomly selected (50 trials) or aligned to a fixed 2-s period after the last stimulus of each stimulation train (50 trials). We also used surrogate spike trains constructed from the same spike trains used in the original analysis, except that in every trial the spike times were randomly shifted +1–3 s. The values from these manipulations were very similar in both cortex and striatum, and hence were pooled together and used for statistical comparison and presentation (Fig. 3e; yellow distribution) as a single "surrogate group". The entire procedure was repeated but using spike trains where the spikes from the 50 ms after each stimulus were artificially removed (Supplementary Fig. 4e–g).

For firing rates during task execution, we first extracted spiking activity from the running periods of the task (i.e., treadmill onset to end of sequence of movements). Intertrials were not analyzed. Trials were divided into nonoverlapping windows of 250 ms. The firing rate was calculated for each window (spike count divided by 0.25) and smoothed with a Gaussian kernel filter with a standard deviation of 750 ms. The position of the forelimb contralateral to the recording site was extracted from video recordings at 100 fps (see below), and Pearson's correlation coefficient between spiking-activity and the position of the forelimb was calculated for each cell.

**Behavioral apparatus and training**. A treadmill for humans (NordicTrack T6.1) was customized with plexiglass walls (50 cm high) to constrain rats to the walkable area on the belt (80 cm long by 20 cm wide). The motor of the treadmill was controlled by a custom-made program (LabVIEW, National Instruments) and a multifunction computer board (NI USB-6353, National Instruments). A line of LEDs illuminated the whole apparatus. The front wall of the treadmill was equipped with a liquid well to deliver drops of sucrose solution (Fig. 4a). A photodetector positioned at 10 cm from the front wall delimited and signaled the entrances of the animal in the so-called goal area in each trial (black circle, Fig. 4a). A warning signal (1.5 kHz, 65 dB) indicated incorrect early entrances in the stop area.

Animals were handled (2 h d$^{-1}$ for 5 days), familiarized to run on the treadmill at increasing speeds and trained to perform the task. During training the treadmill speed was fixed at (22 cm s$^{-1}$, 40–160 trials per session, 1 session per day). Trials started independently of the animal's position, but after a few training sessions rats spent most of the inter-trial periods in the stop area. We established a criterion of performance accuracy, ≥72.5% of correct trials over the last 40 trials, for ≥3 consecutive sessions. During training, the experimenter was not physically present in the behavioral room. To determine the position of the animal we used a CCD camera (acA640-120fc, Basler, 100 frames s$^{-1}$, 9 pixels cm$^{-1}$) positioned laterally to the treadmill and a fluorescent marker attached to the left forelimb. The marker's positions or the center of the body of the animal were automatically extracted with a custom-made program (Vision, National Instruments) and averaged in 400-ms long sliding windows (the average duration of a step cycle). To quantify the stereotypy of behavior, we extracted the position (Fig. 4e) and speed (Fig. 5a) time-courses from every trial of every session. Position (or speed) trajectories were aligned to the entrance times (i.e., end of sequence of movements), and Pearson's correlation coefficients were computed for all the possible pairs of trials (for speed we restricted the analysis to the last 2.5 s before entrance time).

**Pharmacological lesions and lesion evaluation**. Pharmacological lesions were performed by infusing NMDA (Sigma, 200 mM in sterilized saline) in the striatum (0.8 μl; coordinates in mm with respect to bregma: AP = +0.6; DL = 3.6; V = −4), VPL (0.4 μl; AP = −2.3; DL = 2.8; V = − 6.6) or cortex (0.8 μl; AP = + 0.6; DL = 3.6; V = − 1.5). To quantify the size of lesions, anatomical magnetic resonance scans were performed with a Bruker Pharmascan 70/16US, 7 T MR scan (Bruker, Ettlingen, Germany) under Ketamine anesthesia (85 mg/kg). The anatomical scans were acquired using a spin-echo rapid acquisition with refocused echo (Turbo-RARE) sequences with the following parameters: repetition time = 1800 ms; echo time = 38 ms; RARE factor = 16; number of averages = 2; field of view = 18 × 20 mm$^2$; matrix dimension = 144 × 160; slice thickness = 0.5 mm, resulting in voxel size of 0.08 × 0.08 × 0.5 mm$^3$. MRI T2 images were pre-processed with a free code developed by Coupé et al.[63] and Tuistison et al.[64], which applies non-local means denoising using minc-toolkit and the N4 bias field correction from ANTs. After pre-processing, manual segmentation of unilateral lesions was performed with ITK snap to create masks of each region of interest (DLS, S1, and VPL). The lesioned area was identified by hyperintensity voxel change compared with the non-lesioned counterpart. Finally, lesion volume quantifications were performed with Rstudio. After the experiments all animals were sacrificed under deep anesthesia, and their brains were extracted and subjected to standard histological procedures to confirm the size of their lesions.

**Reversible inactivation of the DLS in behaving animals**. Rats under deep sevoflurane anesthesia were bilaterally implanted with guide cannulae in the VPL (coordinates in mm with respect to bregma: AP = − 2.3; DL = ± 2.8; V = −5.6; the injectors protruded the guide-cannulae by 1 mm). Local injections (500 nl per site at 200 nl min$^{-1}$) were performed 15 min before the behavioral sessions. The GABA$_A$ agonist muscimol (Tocris) was diluted in saline and injected at different concentrations (50 ng μl$^{-0.5}$, 100 ng μl$^{-0.5}$, 500 ng μl$^{-0.5}$, and 1 μg μl$^{-0.5}$). The 500-ng and 1-μg doses induced potent behavioral effects; animals presented low muscle tone and general indisposition to move or run. The two lowest doses had no apparent effect on basic locomotor activity in the home cage; hence, we use the 50-ng dose for behavioral evaluation on the treadmill. Every animal received 2–3 injections of the target dose separated by at least 15 days. At the end of experiments animals were perfused under deep anesthesia and their brains were extracted and processed for histological confirmation.

**Viruses and optogenetic manipulations**. Viruses were obtained from the University of North Carolina Vector Core. A total volume of 200–400 nl was injected unilaterally (anesthetized experiments) or bilaterally (behavioral experiments) in the VPL (coordinates in mm with respect to bregma: AP = − 2.3; DL = ± 2.8; V = − 6.6). For the behavioral experiments, in the same surgery optical fibers (200 μm diameter) were implanted bilaterally 500 μm above the site of injection in the VPL or its terminals in the DLS (in mm with respect to bregma: AP = 0.6; DL = ± 3.8; V = − 4). We used 5 different stimulation protocols, continuous stimulation, and train stimulation at 5, 3.3, 2, and 1 Hz. Stimulations were given bilaterally and in the case of trains alternated (i.e., one site was delayed by half a cycle). In all animals, increasing the light intensity produced forelimb movements in rest and

increased exploration. Since these results could affect the basic running capabilities during the task, for each animal we adjusted the intensity to 75% of the minimum necessary to produce movements. We also tested higher frequencies (8 and 10 Hz) and bilateral simultaneous trains; in both cases the animals were unable to move, so these protocols where not used in treadmill sessions. For all optogenetic experiments, we used LED technology. All hardware was acquired from Plexon including, PlexBright LD-1 single channel LED drivers, PlexBright LED modules (Blue, 465 nm; Green, 525 nm; Lime, 550 nm), PlexBright Dual LED commutators, Optical Patch cables, Fiber Stub implants.

**Reporting summary**. Further information on research design is available in the Nature Research Reporting Summary linked to this article.

## Data availability
The data sets generated during and/or analyzed during the current study are available from the corresponding author on reasonable request

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

## Acknowledgements

We thank David Robbe for critical comments, discussions and support. Ana Inácio, Hugo Merchant, and Luis Téllez for critical reading and discussions on this MS. Authors thank the support provided by all the members of Laboratory A-02 from the Institute of Neurobiology, UNAM; Cuautli Pacheco for providing invaluable support in animal maintenance and care and in histology; Oscar Prospéro for generous donations of indispensable equipment; Joel Han, Jorge Larriva and Víctor Vargas for advising on retrograde tracing; Siddhartha Mondragón, Nydia Hernández, and Martín García for assistance in microscopy and animal care; Juan Ortiz and the "Laboratorio Nacional de Imagenología por Resonancia Magnética" for support in magnetic resonance imaging experiments. Anaid Antaramian and Adriana Gonzalez from Unidad de Proteogenómica, INB. Jessica Gonzalez-Norris for proofreading. A.H-B is a master's student supported by fellowship 629356 from CONACyT-Mexico. A.Y.L. and A.P.-F. are PhD students from Programa de Doctorado en Ciencias Biomedicas and supported by fellowships 696111 and 463747 from CONACyT-Mexico. This work was funded by grants UNAM-DGAPA-PAPIIT: IA201916, IA201018 to P.R.-O.; and CONACyT: FDC_1702 to P.R.-O.

## Author contributions

P.R.-O. designed and performed the experiments, analyzed the data, supervised the project, and wrote the paper. A.H.-B. performed electrophysiological experiments on anesthetized and freely moving animals and histological confirmations. A.Y.L. performed the anatomical and electrophysiological experiments on anesthetized animals. A.P.-F. and M.P.-R. performed the magnetic resonance imaging experiments. All authors contributed during training of the animals, revised the article, and approved the final version.

## Additional information

**Competing interests:** The authors declare no competing interests.

