## [Peer Review File · Nature Communications]

Reviewers' Comments:

Reviewer #1:

Remarks to the Author:

This study examines the role of the ventroposterior lateral nucleus in the execution of a basal ganglia-dependent skilled timing behavior in rats. A predominant theory for how timing may be accomplished in the brain argues that sensory input and reinforcement-learning shapes neural networks into generating dynamic activity patterns that can be used as an internal clock. Such clocks have been shown in both cortex and striatum, but the origin of the sensory stimuli that are thought to drive these clocks are unknown. The study presented here investigated the contribution of VPL projections to the striatum to timing behavior. The experiments are well designed and carefully conducted, and the data quality is high. The authors used a number of complementary techniques including in-vivo recording, lesions, pharmacological inactivation, and inhibitory and excitatory optogenetics to provide evidence that this pathway may be involved in mediating the performance of a timing task.

Since the role that thalamic inputs play in patterning and structuring striatal activity is not well understood, this study is timely and of significant interest to the field. I do have a number of significant concerns.

Main concerns:

1. The task contains a timing component that the authors argue can be separated from other kinematic variables, such as speed and stereotypy. . It would be helpful to provide more characterizations of the kinematics, as I find it difficult to understand how the entry time can be consistently lower across all of the manipulations in the study without being accounted for by any changes in kinematics or performance. I would suggest adding an additional supplementary figure to further quantify kinematic variables during the 3 different phases of the task (front, back, and front). Do the animals spend more time in the back phase of the task after the manipulations? Are they spending more time in the front prior to the back phase? For example, it seems from Figure 6E that the representative animal spent less time in the holding phase at the back of the treadmill and that there is more variability in the behavior. This was reported but not quantified. Variability of speed, position, and acceleration should also be quantified and graphed.
2. Population dynamics and decoding (Figure 3). The experiment involves stimulating the pad of the forepaw (I assume it was the pad, it is not stated in the methods) in a rhythmic manner to mimic the sensory stimulations experienced during the behavioral task. Striatal and cortical recordings revealed a rhythmic influence of these stimulations on the neural populations of both areas. The necessity of this analysis and the interpretation of the results is difficult to fit into the rest of the manuscript. I would suggest either removing the decoding component or explaining in more detail how it is relevant. For example, previous work has suggested the striatal dynamics account for the evolution of time during a task, upwards of dozens of seconds. If stimuli coming from rhythmic locomotor activity in the forepaw enter the striatum via the VPL and initiate sequential dynamic activity with each step, then decoding time from the population activity throughout the trial would be nearly impossible, as there would be a repeating sequence with each step. The fact that other groups have extracted time over many seconds suggests that sensory input is at best used a manner that is different from what the authors demonstrated here. These issues should be discussed more extensively.
3. The bidirectional effect of optogenetic manipulation is the major strength of this paper. The optogenetic stimulations were performed beginning with the back phase of the task. I am interested in understanding further what the effects of optogenetic stimulation are on kinematics of the trial (similar question as in point 1, above). For example, if the stimulation using ChR2 increased the duration of the Back phase, then it is trivial to report that these groups had increases in the number of correct trials, as that would be a direct consequence of spending more time in a specific phase of the task. One also naturally wonders what would happen if the stimulations were applied during different

phases of the task, for example during the acceleration phase only. If inhibition of VPL leads to a reduction in the amount of time spent during the Back phase, would it actually cause an increase in speed during the acceleration phase?

Minor issues:

1. Terminal stimulation while recording in the striatum. Could the shorter latencies in these recordings be due to light artifacts from local laser activation? Please provide a control recording with no ChR2 expression.
2. The methods used for Figure 3 and supplementary figure 3 are unclear. There are potentially simpler approaches that could be taken for determining the similarity of trajectories during each of the paw stimulations. It would for example make sense to make an average trajectory from activity over all stimulations in each brain area and then calculate the correlation coefficient with a sliding window moving through the activity recorded during the stimulation train. It was confusing that there were 5 different template trajectories used, yet there is only one trace per brain area in Figure 3B. Please improve the description of the analysis or find an alternative approach. For the population decoding, the methods state that the binary classifiers' scores are reported somewhere but I could not locate where these results are presented. Furthermore, in this plot, there is a horizontal dotted line at 0.5, yet it is not described what this refers to. If I am not mistaken, it might reflect a chance performance threshold, yet chance performance in this analysis would be something closer to 1/60. I would suggest adding a supplementary figure that more clearly describes the algorithm used for decoding and for guiding the reader toward the final results shown in Figure 3E.
3. Please check that all figure labels are correct in the text and legends as there were some issues.
4. Supplementary Figure 2: Please indicate the approximate boundary between S1 and DLS.
5. Several errors in references. Incorrect years are used or the references themselves are incomplete.

Reviewer #2:

Remarks to the Author:

In this manuscript, Luma et al. explore the roles of somatosensory inputs to the striatum in an animal's (rat's) ability to perform a timed treadmill walking task. Through a series of experiments, the authors show that somatosensory thalamus (VPL) projects to the dorsolateral striatum (DLS), that (in anesthetized animals) striatal neurons respond to somatosensory stimulation at latencies similar to those seen in S1, that neural responses evolve across a train of 5 sensory stimuli more in DLS than in S1, and that a variety of circuit manipulations (permanent lesions, muscimol inactivations, and optogenetic manipulations) all selectively interfere with a rat's ability to learn and perform the self-timed component of the timed treadmill walking task. The authors conclude that somatosensory inputs to the DLS (presumably from sensory reafference) "provides the temporal reference for the development and execution of motor habits."

The authors bring to bear an impressive combination of experimental approaches to build a case for the importance of somatosensory inputs to the DLS for an animal's ability to perform behaviors like those required by the treadmill task. The manuscript would benefit, however, from added attention to the following points.

MAJOR COMMENTS

Novelty: The importance of somatosensory inputs to striatum for complex behaviors has been examined previously by several investigators. The authors are encouraged to consider the studies of Romo, Merchant, et al. (Neuroreport 1995 and J Neurophysiol 1997) and those from the lab of RJ Nelson (e.g., Exp Brain Res 1992; Front Neurosci 2011). The work of Samuel Liles is also relevant (J Neurophysiol 1985) although the task used there wouldn't be considered "complex." Many more

recent publications build on those seminal studies.

“somatosensory flow... provides the temporal reference”: This general concept is complicated by the specific effects observed on task behavior. In the first parts of the manuscript the authors build a story that striatal responses to somatosensory inputs may “represent elapsed time” during performance of the treadmill task. The simplest interpretation of this model is that striatal responses correlate positively with elapsed time – essentially that each striatal response moves the “striatal population dynamics” forward in time. The complication is that all of the experimental manipulations that reduced sensory inflow (permanent disconnection, muscimol and opto-silencing) caused rats to respond more quickly than appropriate, as if they over-estimated the passage of time. And manipulations that increased sensory inflow caused rats to respond more slowly than normal, as if they under-estimated the passage of time. It is difficult to see how all of the results fit together into a coherent picture.

Convergent striatal inputs from sensory and motor cortices: The manuscript scarcely mentions motor inputs to the DLS. Though it is understandable that the authors wish to focus on the often underappreciated sensory inputs, the manuscript could give many readers a distorted impression concerning the major inputs to DLS. Recent studies provide strong evidence for convergent input to individual striatal neurons from primary sensory and primary motor cortices (see Hunnicutt et al. eLife 2016 and Hooks et al. Nat Commun 2018). (See also the classic work by Flaherty & Graybiel on somatotopically-specific convergence in the striatum of inputs from S1 and M1.) Given these data, do the authors think it tenable to exclude motor signals from their general thesis that “somatosensory flow ... provides a temporal reference”?

Somatotopy in S1 and DLS: The manuscript does not mention how the locations of injections and observations relate to the known somatotopy in S1 and DLS. Were forelimb related locations targeted? This point is especially important for the experiments performed in anesthetized animals. Perhaps many of the long latency responses shown in Figs. 1-3 are due to recordings being from, e.g., a hindlimb related region whereas sensory stimulation was delivered to the forelimb.

Anatomic accuracy of manipulations: The significance of many of the results depends on the assumption that the manipulations were anatomically accurate. For example, for the comparisons of neuronal activity in S1 and DLS it is important that recordings were obtained from the anatomically connected, and presumably forelimb-related, regions of S1 and DLS. For the experiments using lesions, inactivations, or injections targeting the VPL, did the authors ensure that those manipulations did not impinge on the intralaminar thalamic nuclei (Pf or CL)? Results from those experiments would have significantly different implications if those nearby intralaminar nuclei were involved (see recent work from the Balleine lab). Muscimol and viral vectors can spread a significant distance beyond the site of injection.

SPECIFIC COMMENTS

pg 3 “the forelimb areas of the thalamus” – This phrase could easily be mistaken as a statement that the only forelimb-related area of thalamus is in VPL. And it reinforces the perspective that the motor domain is completely irrelevant. This probably should be reworded to emphasize that VPL is a somatosensory specific thalamic nucleus.

pg 4 “late response between 100 and 300 ms”: These are extremely long latencies likely long enough for conduction of sensory volleys multiple times around any of several loop circuits (cortico-thalamic, cortico-cerebellar, cortico-basal ganglia, etc.). Related to this, the Methods description of how

somatosensory stimulation would benefit from substantially more detail. What part of the forelimb was stimulated? What kind of movement was imposed? Did the experimenters ensure that the mechanical perturbation was localized and not conducted to other parts of the body?

pg 5 "apparat" – spelling

pg 5 "Visual inspection of the data... cortical sequential activation ... more similar" – This feature of results shown in Fig. 3a could be explained in more detail. First, it will not be clear to many readers why neurons in both S1 and DLS appear to be far more responsive (i.e., showing a phasic increase in firing) to stimulus #3 than to any of the other stimuli. Second, it will not be clear to many readers what difference in response similarity is supposed to be seen by comparing S1 and DLS units. It would probably help to provide a few single-unit examples (i.e., spike density functions) of what is meant by "similarity" (and dissimilarity) of responses across stimuli. And then to provide a written description of how this difference is evidence in Fig. 3a.

pg 6 "step times" and "each step" – To be clear, these are not really steps. These are mechanical perturbations of the forelimb.

pg 6 "five obvious clusters" – the stimulus #5 cluster is not visible in the PDF received. The shade of gray used is virtually white.

pg 8 "performing irreversible pharmacological lesions" – Methods for this appear to be missing from the Methods section.

pg 8 "trough" – spelling

pgs 10-11 "Disruption of sensory flow... to the DLS" and "manipulating the sensory flow to the DLS" – To be accurate, experiments that targeted the VPL manipulated all somatosensory flow, including somatosensory input to cortex. In the current version the text throughout these sections is misleading.

pg 11 "infected the VPL bilaterally" – It is not clear why the histologic confirmation of virus injection (Supplemental Fig. 7a) does not show bilateral label.

pg 11 "another group of over-trained animals (n=8)" – The numbers of animals used for sub-experiments is difficult to follow here. Why are results shown for just two animals in Supplementary Fig. 7a-d?

pg 21 "scarified" – spelling

Reviewer #3:

Remarks to the Author:

In this manuscript, Luma et al. report that a temporal component of somatosensory information into the dorsolateral striatum (DLS) contributes to learning to optimize a patterned running behavior. It consists of electrophysiology on sensory responses of the DLS and primary somatosensory cortex (S1) neurons in anesthetized rats (Figs. 1-3) and behavioral observations using task-performing rats with the DLS, S1, and/or thalamus (VPL) lesioned permanently or reversibly (Figs. 4-7). In the anesthetized rats, the authors showed sensory inputs from VPL well explained similar spike responses of the S1 and DLS neurons to tactile stimuli on a forelimb. If stimulated repetitively like a running

state, the S1 neurons responded more reliably to each stimulus than the DLS. In other words, the DLS could convey more temporal information than the other. In the behavioral experiments, the authors utilized their original task in which rats must run into a goal on time on a treadmill (Rueda-Orozco, 2015). They made a functional disconnection among VPL, S1, and DLS by combining two permanent lesions of them in the right and left hemispheres. Such brain lesions, especially VPL-DLS disconnection, impaired the behavioral task learning remarkably. Motor skills of running themselves were preserved well in the lesioned rats. Optogenetic activation of VPL neurons enhanced the task performance. On the basis of these results, the authors concluded that the sensory signal through direct VPL-DLS pathway provides a temporal reference to optimize the running behavior. The authors performed several technically difficult experiments and supplementary experiments very carefully, and analyzed the experimental data with appropriate statistics. The manuscript is well written with a conceptually attractive hypothesis. Thus, this work sounds very interesting to basal ganglia researchers. However, I am not fully convinced of their main conclusion, which is merely one of different interpretations, because they made many assumptions in their experimental conditions. It is desirable to reinforce their conclusions by simpler and more direct experiments that I suggest below.

Major comments:

1) It is possible that the reliability of peak responses to repetitive sensory stimuli (Fig. 3) depends merely on the S/N ratio of the peak response. In general, cortical neurons, in particular in deep layers, show much higher spike activity on average than striatal projection neurons. In example neurons (Fig. 1c), however, S1 neurons display somehow lower baseline activity and prominent peaks, and thus, the S/N ratio looks much better in the S1 neurons than the DLS (Fig. 1d). The peak activity with higher S/N ratio means less variance (jitter) in the peak time. It can explain more reliable responses to the repetitive stimulation in the S1 neurons, which does not support their temporal reference hypothesis.

2) The disconnection method, lesioning one area and other area contralaterally (Fig. 4b), is very complicated and tricky, although it was reported previously. In this method, the disconnection is defined as a symmetric abolishment of the same interareal connections on both sides. However, even though other connections are also damaged on only one side, they are considered "functional" whether ipsi- or contralateral, ignoring any effect of such unilateral damages. For example, the VPL-DLS disconnection is actually accompanied by left S1-DLS disconnection and right VPL-S1 disconnection. Its assumption also ignores compensatory changes or side effects in the three and other (e.g., callosal) connections. In addition, the difference between the VPL-DLS disconnection and Partial disconnection (as a control) could be regarded as the presence/absence of an intact loop among the three areas on one side. Taken together, the VPL-DLS disconnection cannot be truly pathway-specific disconnection. This point should be clearly described in the manuscript.

3) It is, therefore, necessary to show statistical comparisons of sensory spike responses of DLS neurons among these disconnection conditions in Sup. Fig. 2. More importantly, impaired sensory signals in the DLS neurons should be confirmed electrophysiologically during the task performance in those lesioned groups.

4) I would suggest simpler and more direct experiments using optogenetic inhibition to block the pathway signal specifically. Briefly, Halorhodopsin or Archaelrhodopsin-T is expressed in the VPL neurons bilaterally by viral vector infection, and their axonal terminals in the DLS are inactivated by local light illumination during task performance, which allows to inhibit the VPL-DLS pathway specifically. The optogenetic activation of VPL-DLS terminals with Channelrhodopsin-2 (Fig. 7e-h) will propagate back into other areas through axonal collaterals and can affect behavioral performance indirectly. Thus, it is not direct evidence of the pathway-specific function.

5) The S1 looks more damaged than the DLS in the left MRI image of Sup. Fig. 4a. Sup. Fig. 4d shows certainly similar lesioned areas in the S1 and DLS. An additional analysis should be performed using only individuals with larger DLS lesion and less S1 lesion.

6) Fig. 7 and Sup. Fig. 7. Did the repetitive stimulation at 3.3 or 5 Hz indeed affect the frequency (Hz) of locomotion cycles as a temporal reference? If it improved the task score (entry time) without changing the locomotion cycles, what does the repetitive sensory responses mean in Sup. Fig. 2?

Minor comments:

- 1) Fig. 1 legend (p.23). Panel letters "b" to "h" for Fig. 1 panels were shifted in the legend.
- 2) Fig. 1a and Sup. Fig. 1b-d. FG-positive should be unified to either lighter (white, fluorescence) or darker (black, reversed).
- 3) Fig. 2a, Fig. 7a,e, and Sup. Fig. 7a,f. Font size is too small. Is each photo a combination of two photos on the right and left sides? If so, they should be separated with a gap between them or by labeling. Also, the direction should be indicated by R and L or M and L on these photos.
- 4) Fig. 2d. Gray for mechanic is too faint to see. It should be darker.
- 5) Fig. 3a-c. I cannot see "stim. 5" too, which should be darker.
- 6) "VPL-DLS disconnection", etc. are misleading as I mentioned above. I would suggest more correct wording, for example, "cVPL & DLS lesion" (c, contralateral).
- 7) Sup. Fig. 4a,b,c. The lesioned areas should be marked by arrows or arrowheads.
- 8) Sup. Fig. 2. I would prefer to move Sup. Fig. 2 behind Sup. Fig. 4.

Reviewer #1:

Main concerns:

1. The task contains a timing component that the authors argue can be separated from other kinematic variables, such as speed and stereotypy. It would be helpful to provide more characterizations of the kinematics, as I find it difficult to understand how the entry time can be consistently lower across all of the manipulations in the study without being accounted for by any changes in kinematics or performance. I would suggest adding an additional supplementary figure to further quantify kinematic variables during the 3 different phases of the task (front, back, and front). Do the animals spend more time in the back phase of the task after the manipulations? Are they spending more time in the front prior to the back phase? For example, it seems from Figure 6E that the representative animal spent less time in the holding phase at the back of the treadmill and that there is more variability in the behavior. This was reported but not quantified. Variability of speed, position, and acceleration should also be quantified and graphed.

We thank the reviewer for this comment; indeed, the work would significantly improve by quantification of the times and variabilities spent in each phase of the sequence. We have performed these analyses for the learning curves of the animals trained under different lesions and pharmacological and optogenetic manipulations. The results confirm that the main effects of the treatments are in the amount of time spent in the “holding phase” on back of the treadmill. These results further indicate that manipulating the sensory inputs to the DLS specifically disrupts the part of the motor sequences that is more susceptible to temporal components. The new analyses are included in two supplementary figures (Supplementary Fig. 8 & Supplementary Fig. 11) and explained in the main text (Page 10 line 5 to 14).

2. Population dynamics and decoding (Figure 3). The experiment involves stimulating the pad of the forepaw (I assume it was the pad, it is not stated in the methods) in a rhythmic manner to mimic the sensory stimulations experienced during the behavioral task. Striatal and cortical recordings revealed a rhythmic influence of these stimulations on the neural populations of both areas. The necessity of this analysis and the interpretation of the results is difficult to fit into the rest of the manuscript. I would suggest either removing the decoding component or explaining in more detail how it is relevant. For example, previous work has suggested the striatal dynamics account for the evolution of time during a task, upwards of dozens of seconds. If stimuli coming from rhythmic locomotor activity in the forepaw enter the striatum via the VPL and initiate sequential dynamic activity with each step, then decoding time from the population activity throughout the trial would be nearly impossible, as there would be a repeating sequence with each step. The fact that other groups have extracted time over many seconds suggests that sensory input is at best used a

manner that is different from what the authors demonstrated here. These issues should be discussed more extensively.

We apologize for the lack of clarity; stimulations were given in the pads of the forepaw contralateral to the recording site. We have clarified this point in the Methods section (Page 21 line 48 to page 22 line 9). Also, following a suggestion from one of the reviewers, we have included a new supplementary figure schematizing this point (Supplementary Fig. 2).

Regarding the remaining part of this comment, we agree with the reviewer; indeed, if stimuli entered the DLS and triggered very similar sequential activations in each stimulus and every train, it would be impossible to decode time. On the contrary, if each stimulus of the train evoked a “different” sequence in every stimulus of a train, but stable over different trains, then it would be possible to decode time. This is precisely what we found in the DLS. When looking at the population responses, we found that each stimulus of the sequence produced a distinguishable “network state” that was very stable across trains. In other words, for the population dynamics in the DLS, the stimulus number 1 for all trains evoked activity that ended in a particular position of the population space, and this position was very different to the position where all stimuli 2, 3, 4 and 5 landed (original Fig. 3c, now Fig.3b). Under our anesthetized conditions, these data are consistent with repetitive sensory stimulations providing signals from which time could be decoded at least in the scale of various hundreds of milliseconds, the temporal domain for the execution of single motor sequences like the ones presented in this work. The data are also consistent with the previous report from Bakhurin et al. (2017), where time was decoded from striatal population activity in the temporal domains of ~2.5 s.

The possibility that rhythmic sensory inputs could sustain temporal representations in the scale of dozens of seconds to minutes would need to be addressed experimentally. It's important to notice that in studies where temporal representations in the DLS occur for such longer timescales (e.g. Mello et al 2015), the animals were not necessarily engaged in a particular sequence of movements nor did they develop stereotypical movements; hence, it is not possible to assume the presence of sustained periodical sensory inputs to the DLS. Our data only support the idea that sensory information is fundamental for the execution of timed sequences of movements that are generally presented in shorter periods of time.

We apologize again to Reviewer #2. This point should have been more clearly developed. We have improved the Results and Methods sections accordingly. With these clarifications we hope that the Reviewer will find the results sufficiently relevant to accompany the rest of the publication.

3. The bidirectional effect of optogenetic manipulation is the major strength of this paper. The optogenetic stimulations were performed beginning with the back phase of the task. I am interested in understanding further what the effects of optogenetic stimulation are on kinematics of the trial (similar question as in point 1, above). For example, if the stimulation using ChR2 increased the duration of the Back phase, then it is trivial to report that these groups had increases in the number of correct trials, as that would be a direct consequence of spending more time in a specific phase of the task. One also naturally wonders what would happen

if the stimulations were applied during different phases of the task, for example during the acceleration phase only. If inhibition of VPL leads to a reduction in the amount of time spent during the Back phase, would it actually cause an increase in speed during the acceleration phase?

We agree; the bidirectional effect of optogenetic manipulation is a strong component of the paper. Given the importance of this comment and a suggestion from another reviewer, we have included two more animals in which the VPL terminals were inhibited in the DLS during the execution of the task. For these reasons, we have modified Figure 7, originally dedicated to optogenetic stimulation, now include inhibition (Fig. 7 a-d) and excitation (Fig. 7 e-f) of the VPL and its terminals in the DLS. The new data further support the bidirectional effect.

For the second part of the comment: ***“The optogenetic stimulations were performed beginning with the back phase of the task. I am interested in understanding further what the effects of optogenetic stimulation are on kinematics of the trial (similar question as in point 1, above)”***

Regarding point 1, we now report the analysis of the amount of time spent in the different phases of the sequence (Page 13 line 6 to 18) in Supplementary Figure 11. After this analysis, the main observation is that the increases (Channelrhodopsin-II animals) and decreases (Halorhodopsin/Archaeorhodopsin animals) in entrance times can be explained by the animals spending more (or less) time at the back of the treadmill, specifically during the “holding phase”.

For the third part of the comment, ***“if the stimulation using ChR2 increased the duration of the Back phase, then it is trivial to report that these groups had increases in the number of correct trials, as that would be a direct consequence of spending more time in a specific phase of the task”***.

Is also correct, but while the increase in amount of rewarded trials is a direct consequence of the animals spending more time at the back of the treadmill, we believe is important to show that the “time shifts” have a direct impact on the final outcome of the execution. In response to the reviewer’s comment, we have modified the figure and now we only show one example of the percentage of correct times for each of the four conditions, stimulation/inhibition on VPL or its terminals (Fig. 7 c and g).

For the fourth part of the comment, ***“One also naturally wonders what would happen if the stimulations were applied during different phases of the task, for example during the acceleration phase only”***.

A partial response to this question is already in the original version of the MS. If the main effects of stimulation occur during the holding phase of the sequence, then we could expect little or no effects when stimulating in the acceleration phase only. This can be seen in our original experimental design. In every stimulated trial, stimulation started

when the animals reached the back of the treadmill and was maintained until the animals finished the sequences of movements including the last acceleration. The data presented in the original MS demonstrate that the last part of the sequence was not affected by the optogenetic manipulations, even when stimulation was still present when the animals were executing this phase (Fig. 7d and h). This was also confirmed by the new analysis proposed by the reviewer, where the time spent in this phase is not significantly change (Supplementary Fig 11). Nevertheless, definitive proof can be obtained experimentally; for this purpose, we infected two new well-trained animals with Channelrhodopsin 2 in the VPL. These animals were also used to perform simultaneous optogenetic stimulations and silicon probe recordings while executing the sequence. After recordings were completed, we performed stimulations in the last phase of the sequence. The new data is consistent with the previous results, as stimulation only in this phase did not alter arrival times or kinematic parameters. This is now reported in Supplementary Figure 10 b-d and explained in the Results section (Page 13 lines 6 to 18).

Finally, for the last part of the comments, ***“If inhibition of VPL leads to a reduction in the amount of time spent during the Back phase, would it actually cause an increase in speed during the acceleration phase?”***

The answer is no, the stimulation did not cause an increase in speed in the acceleration phase in the four animals under optogenetic inhibition (Supplementary Fig. 11 a and c).

Altogether, this set of new results and analysis further support the notion that manipulating the sensory input to the DLS during the execution of the sequence mainly affects the temporal component of the execution.

Minor issues:

1. Terminal stimulation while recording in the striatum. Could the shorter latencies in these recordings be due to light artifacts from local laser activation? Please provide a control recording with no ChR2 expression.

Indeed, this could be a possibility; we have performed the experiments proposed by reviewer. It is important to clarify that we did not use laser but rather LED technology for all our original and new experiments. We apologize for not clearly stated this in the original MS, now this information is included in the Methods section (Page 25 lines 20-23). For these new experiments we recorded 404 cortical cells and 163 striatal cells from 3 non-infected animals (Supplementary Fig. 3 a-c). Like the cells reported in the original version of the paper, the new groups of cells presented robust representations of mechanical stimulations of the contralateral forelimb. Importantly, none of the recorded cells presented responses to direct illumination of the cortex or striatum with blue light applied with the same parameters as those in the previous experiments. The new set of data is included in the new Supplementary Figure 3 and explained in the main text (Page 4, line 31 to page 5 line 3).

2. The methods used for Figure 3 and supplementary figure 3 are unclear. There are potentially simpler approaches that could be taken for determining the similarity of trajectories during each of the paw stimulations. It would for example make sense to make an average trajectory from activity over all stimulations in each brain area and then calculate the correlation coefficient with a sliding window moving through the activity recorded during the stimulation train. It was confusing that there were 5 different template trajectories used, yet there is only one trace per brain area in Figure 3B. Please improve the description of the analysis or find an alternative approach. For the population decoding, the methods state that the binary classifiers' scores are reported somewhere but I could not locate where these results are presented. Furthermore, in this plot, there is a horizontal dotted line at 0.5, yet it is not described what this refers to. If I am not mistaken, it might reflect a chance performance threshold, yet chance performance in this analysis would be something closer to 1/60. I would suggest adding a supplementary figure that more clearly describes the algorithm used for decoding and for guiding the reader toward the final results shown in Figure 3E.

We followed the reviewer's advice; we have substituted our original analysis for the proposed one. The results of the new analysis are consistent with our previous results; that is, cortical population trajectories evoked by single stimulus are more similar between each other than striatal ones. The old analysis was removed from the paper and the new analysis is now included in the new Supplementary Fig. 4a.

For the second part of the comment, we have included the missing descriptions in the figure. We apologize for this imprecision; what is actually reported is the median + 25th and 75th percentiles of the 500 iterations with randomly selected pools of cells (Figure 3 e). The last value of this plot corresponds to the median + 25th and 75th percentiles of the 500 iterations with the full sample of cells.

We plotted a dotted line as graphic visual reference. In this case it does not indicate the random value. We extracted this value by creating confidence intervals running the same analysis in (1) randomly selected 3-s periods (50 trials) where no stimulation was provided; (2) 3-s periods aligned to a fixed 2-s period after the last stimulus of each stimulation train (50 trials); and (3) surrogate spike trains constructed from the same spike trains used in the original analysis, except that in every trial the spike times were randomly shifted $\pm 1-3$ s. When performing these manipulations, the predictive values from S1 and DLS were very similar and thus pooled together.

Following the reviewer's advice, we have also included a schematic view of the procedure (Supplementary Fig. 4d) and improved the description in the Results and Methods sections.

3. Please check that all figure labels are correct in the text and legends as there were some issues.

We have revised the manuscript and corrected this point.

4. Supplementary Figure 2: Please indicate the approximate boundary between S1 and DLS.

We have included this boundary.

5. Several errors in references. Incorrect years are used or the references themselves are incomplete.

We have revised and corrected these problems; we apologize for these errors.

Reviewer #2 (Remarks to the Author):

MAJOR COMMENTS

Novelty: The importance of somatosensory inputs to striatum for complex behaviors has been examined previously by several investigators. The authors are encouraged to consider the studies of Romo, Merchant, et al. (Neuroreport 1995 and J Neurophysiol 1997) and those from the lab of RJ Nelson (e.g., Exp Brain Res 1992; Front Neurosci 2011). The work of Samuel Liles is also relevant (J Neurophysiol 1985) although the task used there wouldn't be considered "complex." Many more recent publications build on those seminal studies.

Regarding novelty, we did not intend to suggest that this is the first paper examining the importance of somatosensory inputs to the striatum in complex behavior. Nevertheless, as stated by the reviewer in his/her following comment, sensory inputs to the DLS are often under-appreciated. As a result, we still don't have a clear idea of how the striatal microcircuits process sensory information, and even less of how sensory representations in the DLS are used to guide behavior. Given the nature of our experiments, we focus our theoretical framework on references that were relevant in rodent literature and indeed left out many others, especially from primate research. We thank the reviewer for this observation. We did not mean to diminish these important contributions; in fact, some of the investigators suggested by the reviewer were constantly or at some point consulted while making this manuscript and the project itself. With that said, here are the main differences between the present work and the suggested literature:

1. In the studies of Merchant & Romo (1995 & 1997), the main objective was to understand the role of the putamen in sensory discriminations, somesthetic perception and decision making. Animals were trained to categorize their decisions between fast or slow trials based on the speed of mechanic vibrating stimulations

in the finger tips. Consistently with the sensorimotor function of this region, the analysis of neural responses revealed groups of cells that were related to the stimulus (sensory cells) or the movement (motor cells). Interestingly, they also found a group of cells related to the category of the stimulation, i.e. if the stimulation was considered fast or slow. The authors concluded that, given the presence of “category signals”, the putamen may be implicated in decision making. In this sense, these papers can be better contextualized in the large body of literature studying the role of the basal ganglia in decision making and action selection. While this is an important contribution, the authors did not discuss or experimentally address the role of sensation in the actual execution of behavior or how the sensory (or motor) signals help to build the representation of “category”.

2. In the studies by the group of Nelson, the objective was to compare if visual and somesthetic stimulation would produce qualitatively different responses in the DLS of monkeys trained to make wrist flexions or extensions when instructed by visual or vibration cues. Recordings in the putamen and caudate nucleus revealed that both types of stimulations elicited similar changes in the majority of the cells, being the most prominent a pre-movement activation in the range of hundreds of milliseconds. These pre-activations appeared to be independent of the sensory modality of the instruction. The authors discuss that these activations are associated with the initiation of actions. Their work can be considered as one of the earliest but many demonstrations of the implication of the BG in action selection. Even when the relevance of this study is obvious, like Merchant & Romo, the authors did not explore the specific role of sensation in movement execution.
3. The paper of Lilles (1985) is an important piece of evidence where the author demonstrated by recording single cells in the putamen of trained primates, that ~40% of the cells with wrist movement-related signals are also modulated by passive displacement of the same joints. If we interpret the passive displacement of joints as sensory stimulation, this paper opens the possibility of a direct influence of sensory information over motor commands. The study, however, did not address those possible interactions and their potential outcomes in specific behaviors.
4. All in all, from the suggested literature we know that sensory signals are represented in the putamen of primates, and those sensory signals may be useful for action selection and decision making. We also know that a group of cells may integrate both sensory and motor signals. These authors, along with many others including Rueda-Orozco & Robbe (2015), reported the coexistence of sensory signals with non-sensory and many times complex representations, such as speed, position, time and initiation/termination of actions. Nevertheless, none of the articles directly analyzed the specific role of striatal sensory representations in the actual execution of movements (even less automatized sequences of movements) or in the construction of other representations such as speed or time.
5. Another important difference between the present work and those suggested by the Reviewer is that sensory information was inferred by the correlations with spiking activity of individual cells in behaving animals. In our study, even when using an anesthetized preparation, we evaluated sensory information in both individual cells and population dynamics. This approach has revealed a direct link between sensory information and temporal-related properties in the basic

dynamics of the DLS in the absence motor commands. Based on this observation, we designed a set of experiments to causally link sensory information with particular components of motor execution.

6. Furthermore, in the suggested literature the authors did not investigate the origins of sensory representations in the DLS. Here, by using the anatomical, electrophysiological, pharmacological and optogenetic approaches, we have demonstrated the presence of a direct VPL-DLS pathway that significantly influences behavior when manipulated.
7. Finally, the behavioral protocols are fundamentally different and hence reflect a fundamental difference in the objectives of the studies. In the examples suggested by the Reviewer, the animals were trained to make a decision between two choices. The success of the animals depended mainly on their ability to discriminate between different qualities of sensory stimulation and not on the precise execution of the movement. In our behavioral protocol, no particular sensory discrimination was required. The success of the animals depended mainly on their ability to timely execute a motor sequence. While the objective of the suggested studies was to determine the contribution of the putamen to the process of sensory discrimination, perception and decision making, our objective was to determine the role of sensory information in striatal basic computations and its relationship with the actual execution of motor sequences. To our knowledge, our study is the first to explicitly address the role of a particular sensory modality in the population dynamics of the sensory motor striatum and the associated timed execution of an automatized motor sequence.

We have included relevant points of this argumentation in the Discussion in the new version of the manuscript (Page 15, lines 15 to 24).

“somatosensory flow... provides the temporal reference”: This general concept is complicated by the specific effects observed on task behavior. In the first parts of the manuscript the authors build a story that striatal responses to somatosensory inputs may “represent elapsed time” during performance of the treadmill task. The simplest interpretation of this model is that striatal responses correlate positively with elapsed time – essentially that each striatal response moves the “striatal population dynamics” forward in time. The complication is that all of the experimental manipulations that reduced sensory inflow (permanent disconnection, muscimol and opto-silencing) caused rats to respond more quickly than appropriate, as if they over-estimated the passage of time. And manipulations the increased sensory inflow caused rats to respond more slowly than normal, as if they under-estimated the passage of time. It is difficult to see how all of the results fit together into a coherent picture.

We are not in a position to assume that there will be a positive correlation between the population striatal responses and elapsed time; previous literature indicates that population state transitions (like the ones showed in Fig. 3 b) could better reflect the passage of time than simple correlations (Mello, Soares & Paton, 2015). In fact, in previous studies analyzing spiking activity of individual cells (Rueda-Orozco & Robbe,

2015), the number of cells linearly correlated with time was significantly lower than the number of cells correlated with speed or position. Moreover, in the same study, the majority of cells correlated with time presented negative correlations (See Fig. 4 d in Rueda-Orozco & Robbe, 2015). An alternative interpretation is that time is represented in “population clocks” where specific moments are represented as particular network states (Buonomano, 2014). A dynamic like this would be useful to estimate elapsed time and determine a specific stimulus on a train of stimuli (Figure 3 c & e).

In our interpretation, the population sensory representations may serve as a “metronome” or reference for the motor commands. This would be supported by the fact that stimulating or inhibiting sensory inputs to the DLS did not produce significant changes in final accelerations or prevent the animals from following the speed of the treadmill (Figure 7), but only increased or decreased the amount of time spent at the back of the treadmill (Supplementary Fig. 11). This interpretation would also be in line with previous experimental data in primates, which were trained to estimate time intervals in a synchronization-continuation task (Merchant et al, 2011). In the task, monkeys are trained to produce motor commands (by tapping on a button) synchronized to an auditory “metronome” for 3 cycles with a fixed interval; for example, 450 ms. After the 3 cycles of synchronization, the animals are required to produce 3 more cycles with the same interval but in the absence of the auditory cues, i.e. no metronome is present. The authors report a “constant error” defined as a constant shortening of the intervals produced by the monkeys in the absence of the sensory cues. In other words, as the sensory cues disappears, animals tend to “respond more quickly” and produce faster intervals. The error is more noticeable when the intervals to produce are longer, in the scale of ~1000 ms. Interestingly, in this time scale animals are required to produce 6 intervals (3 synchronization + 3 continuation = ~6 s), which is similar to the intervals used in the present task for rodents (7 sec).

We thank the reviewer for this important comment. We have included this reference and arguments in our revised discussion (Page 17, lines 13-21).

Convergent striatal inputs from sensory and motor cortices: The manuscript scarcely mentions motor inputs to the DLS. Though it is understandable that the authors wish to focus on the often-underappreciated sensory inputs, the manuscript could give many readers a distorted impression concerning the major inputs to DLS. Recent studies provide strong evidence for convergent input to individual striatal neurons from primary sensory and primary motor cortices (see Hunnicutt et al. eLife 2016 and Hooks et al. Nat Commun 2018). (See also the classic work by Flaherty & Graybiel on somatotopically-specific convergence in the striatum of inputs from S1 and M1.) Given these data, do the authors think it tenable to exclude motor signals from their general thesis that “somatosensory flow ... provides a temporal reference”?

Indeed, we mainly focused the introduction and discussion on the literature related to sensory inputs to the DLS. In the last section of the discussion we stated “to fully understand the integrative functions of the DLS, it would be necessary to clarify the

interactions with other sensory representations such as the whisker (J. B. Smith et al., 2012) or visual (Reig & Silberberg, 2014) systems, and importantly the with motor-related inputs". As mentioned by Reviewer #2, this is a scarce description of the also very important and most commonly studied motor region of the cortex to the DLS. We purposely focus on sensory regions and projections for the reasons the Reviewer mentions, but we did not intend to confuse the audience with a distorted impression that the major inputs to the DLS are the sensory ones. We have modified the Introduction and Discussion, and now it is explicit that even when the paper is concerned with the sensory inputs to the DLS, this does not mean that this projection is the only or most important input to this region (Page 17 line 28 to page 18 line 4). It is worth mentioning that, as the reviewer acknowledges, the sensory inputs to the DLS are often-underappreciated, giving in many occasions the distorted impression that M1 is the main or most important input to the DLS.

Regarding the second part of the comment, with our current data we are not in a position to exclude (or include) motor signals as significant contributors for the temporal reference thesis. Our data in anesthetized animals indicate that at least sensory inputs would be sufficient to provide this reference; this is also confirmed by the manipulation experiments during the execution of behavior. Another important issue is the structure of the inputs arriving at the DLS, where S1 and VPL would be more suitable to provide a periodic signal (similar to a metronome) than M1 when animals are running on the treadmill. With that said, this important question (M1 vs. S1/VPL) can only be resolved experimentally, which we are already working on but will address in following publications.

Somatotopy in S1 and DLS: The manuscript does not mention how the locations of injections and observations relate to the known somatotopy in S1 and DLS. Were forelimb related locations targeted? This point is especially important for the experiments performed in anesthetized animals. Perhaps many of the long latency responses shown in Figs. 1-3 are due to recordings being from, e.g., a hindlimb related region whereas sensory stimulation was delivered to the forelimb.

Our study is entirely focus on the forelimb region of the S1 and DLS. The stereotaxic coordinates for recordings and injections were selected based on previous literature indicating that these regions in S1, VPL and DLS contain principal representations of the forelimb. Moreover, before starting any experiment, and once having the recording probes in the brain, we inspected the local field potential response evoked by the forelimb and hindlimb stimulations, confirming that all our recordings were performed in the forelimb region. To clarify this crucial point, we have now included explicit statements in the Introduction and Methods and a detailed schematic representation of the recording coordinates as well as LFP examples in a new Supplementary Fig 2.

Regarding the long latencies of some cells, these responses are most likely related to local circuit dynamics previously described and referred to by others as "packets of information" (Luczak et al, 2015). In sensory systems it is common to find this type of responses that spans for hundreds of milliseconds after a single sensory input (Chapin

et al, 1981; Luczak et al, 2009; Bermudez-Contreras et al, 2013). It is worth mentioning that this type of responses has been almost exclusively reported in the cortex, and only recently similar responses were reported in the striatum with single cell techniques (Sippy et al, 2015).

Anatomic accuracy of manipulations: The significance of many of the results depends on the assumption that the manipulations were anatomically accurate. For example, for the comparisons of neuronal activity in S1 and DLS it is important that recordings were obtained from the anatomically connected, and presumably forelimb-related, regions of S1 and DLS. For the experiments using lesions, inactivations, or injections targeting the VPL, did the authors ensure that those manipulations did not impinge on the intralaminar thalamic nuclei (Pf or CL)? Results from those experiments would have significantly different implications if those nearby intralaminar nuclei were involved (see recent work from the Balleine lab). Muscimol and viral vectors can spread a significant distance beyond the site of injection.

We performed electrophysiological recordings in the forelimb region of S1 between layers 4 and 5. The approximate coordinates of these recordings are AP= 0.2 ± 0.3 mm; DL = $3.7 + 0.7$ mm. The microelectrode arrangements we used covered 1.4 mm in the DL axis, and all our electrodes were situated in what was previously described as the forelimb region of S1. Recordings in the DLS were performed under this cortex, a region also described as containing forelimb representations (West et al, 1990; Carelli & West 1991; Rueda-Orozco & Robbe, 2015). Moreover, our retrograde tracer injections in the DLS indicated a direct projection from both S1 (in the coordinates previously mentioned) and the ventral region of the VPL, previously described as the forelimb region of the VPL (Francis, Xu, Chapin, 2008). These results were further confirmed by our electrophysiological and optogenetic experiments in the anesthetized preparation (Figures 1 & 2). All subsequent manipulations in freely moving animals were performed in these cortical, thalamic and striatal regions.

For the second part of the comment, ***“For the experiments using lesions, inactivation’s, or injections targeting the VPL”***. The volumes of injection were calculated to avoid as much as possible spreading to regions outside the target zone. In the case of lesion experiments, we never saw that thalamic lesions would spread to the CL/CM complex. These nuclei can be located in the same anterior-posterior regions. We did not detect lesions in the Pf nucleus, which is around 1 mm away in the AP plane and 1 mm in the DL plane. In case of viral vectors, we did not detect fluorescent signal in the CL/CM nucleus (see Figures 2, 7 and the new Supplementary Fig. 3). Nevertheless, in the case of the Pf nucleus, we typically found stained regions that were more consistent with a VPL-PF projection. In the original version of the MS, we included a section in the Discussion where we stated that the latencies found in optogenetic and mechanical stimulation are more consistent with a direct projection from the VPL rather than an indirect projection; for example, VPL–PF–DLS. We have included a new Supplementary Figure 3 where we show the fluorescence detected in the Pf and also modified the main text to be more explicit in this aspect.

SPECIFIC COMMENTS

pg 3 “the forelimb areas of the thalamus” – This phrase could easily be mistaken as a statement that the only forelimb-related area of thalamus is in VPL. And it reinforces the perspective that the motor domain is completely irrelevant. This probably should be reworded to emphasize that VPL is a somatosensory specific thalamic nucleus.

We have edited this sentence and now we specify that the VPL is a somatosensory region of the thalamus. It is worth mentioning that this point was explicitly stated in the following paragraph of the same page of in the Results section, where we stated the following:

“but also from the VPL (Erro et al., 2002; Erro et al., 2001), implicated in cutaneous and proprioceptive representations of limbs and body (Francis, Xu, & Chapin, 2008)”

pg 4 “late response between 100 and 300 ms”: These are extremely long latencies likely long enough for conduction of sensory volleys multiple times around any of several loop circuits (cortico-thalamic, cortico-cerebellar, cortico-basal ganglia, etc.). Related to this, the Methods description of how somatosensory stimulation would benefit from substantially more detail. What part of the forelimb was stimulated? What kind of movement was imposed? Did the experimenters ensure that the mechanical perturbation was localized and not conducted to other parts of the body?

As mentioned before, in cortical sensory regions it is common to find this type of responses that span for hundreds of milliseconds after a single sensory input (Chapin et al, 1981; Luczak et al, 2009; Bermudez-Contreras et al, 2013). On the other hand, our mechanical stimulations did not produce significant movements that would propagate to other body parts; nevertheless, we also compared mechanical stimulation with direct electrical stimulation of the pads. The responses obtained in both conditions are highly similar. We have included more details about the stimulation conditions in the “Anesthetized experiments” section of Methods (Page 21 line 32 to page 22 line11) and also included a new Supplementary Figure 2 where we compared both types of stimulations.

pg 5 “apparat” – spelling

Corrected.

pg 5 “Visual inspection of the data... cortical sequential activation ... more similar” – This feature of results shown in Fig. 3a could be explained in more detail. First, it will not be clear to many readers why neurons in both S1 and DLS appear to be far more responsive (i.e., showing a phasic increase in firing) to stimulus #3 than to any of the other stimuli. Second, it will not be clear to many readers what difference in response similarity is supposed to be seen by

comparing S1 and DLS units. It would probably help to provide a few single-unit examples (i.e., spike density functions) of what is meant by “similarity” (and dissimilarity) of responses across stimuli. And then to provide a written description of how this difference is evidence in Fig. 3a.

The presentation of these data was unclear for Reviewers #1 and #2, hence we have modified Figure 3. The intention of aligning spiking activity to every stimulus of the train was to visually contrast the striatal and cortical responses to every stimulus of the train. We chose the 3rd stimulus (but showed all in the original Supplementary Figure 4) because in our opinion the difference between cortex and striatum in this stimulus was more evident. Nevertheless, to avoid further confusion we are now only presenting the more traditional alignment to the first stimulus of the train. The formal quantifications of similarity were exposed and statistically compared in the original Figure 3b and the method for these quantifications was sketched in original Supplementary Figure 4b.

Following Reviewer #1’s suggestion and the concerns from Reviewer #2, we have implemented a simpler, more straightforward method to quantify the similarities in the population trajectories evoked by each stimulus of the train.

pg 6 “step times” and “each step” – To be clear, these are not really steps. These are mechanical perturbations of the forelimb.

We have substituted steps by stimuli.

pg 6 “five obvious clusters” – the stimulus #5 cluster is not visible in the PDF received. The shade of gray used is virtually white.

We have changed the color combination to increase the contrast.

pg 8 “performing irreversible pharmacological lesions” – Methods for this appear to be missing from the Methods section.

We have now included this information in the section “Pharmacological lesions and lesion evaluation” (Page 24 lines 24 to 41) in Methods.

pg 8 “trough” – spelling

Corrected.

pgs 10-11 “Disruption of sensory flow... to the DLS” and “manipulating the sensory flow to the DLS” – To be accurate, experiments that targeted the VPL manipulated all somatosensory flow, including somatosensory input to cortex. In the current version the text throughout these sections is misleading.

We have modified the headings and the section itself clarifying this point.

pg 11 “infected the VPL bilaterally” – It is not clear why the histologic

confirmation of virus injection (Supplemental Fig. 7a) does not show bilateral label.

We have corrected this mistake; we showed an image of one of the anesthetized experiments where only one side was infected.

pg 11 “another group of over-trained animals (n=8” – The numbers of animals used for sub-experiments is difficult to follow here. Why are results shown for just two animals in Supplementary Fig. 7a-d?

We apologize for the lack of clarity. In the original version of the paper we used 8 animals for the optogenetic experiments. Every animal is presented individually in either Figure 7 or Supplementary Figure 7. Channelrhodopsin 2, rats A, B, C, D, E in Figure 7 and H in Supplementary Figure 7f-i. Halorhodopsin animals, rats F and G in Supplementary Figure 7a-d. In the new version of the paper, with the inclusion of more animals, we changed this number and edited Figure 7. To avoid confusion, we have removed the general number at the beginning of the section and only kept the numbers of animals for each experiment.

pg 21 “scarified” – spelling

Corrected.

Reviewer #3 (Remarks to the Author):

Major comments:

1) It is possible that the reliability of peak responses to repetitive sensory stimuli (Fig. 3) depends merely on the S/N ratio of the peak response. In general, cortical neurons, in particular in deep layers, show much higher spike activity on average than striatal projection neurons. In example neurons (Fig. 1c), however, S1 neurons display somehow lower baseline activity and prominent peaks, and thus, the S/N ratio looks much better in the S1 neurons than the DLS (Fig. 1d). The peak activity with higher S/N ratio means less variance (jitter) in the peak time. It can explain more reliable responses to the repetitive stimulation in the S1 neurons, which does not support their temporal reference hypothesis.

In principle, the rationale of Reviewer #3 is correct, the reliability of sequential activation may depend on the signal-to-noise ratio response to each stimulus of the train, but the following considerations indicate that it is not the case in this set of data.

First, the firing rates of our cortical cells was 3.79 Hz (percentiles, 75th = 7.16 Hz; 25th = 1.94 Hz), which is very similar to what was previously reported for sensory cortices in recordings performed in rats under anesthesia with urethane (Luzsack et al, 2009, Bermudez-Contreras et al, 2013) and halothane (Chapin et al, 1981). Normally under

these conditions, cortical firing rates are significantly lower than under freely moving conditions; for example, in our new freely moving recordings, average firing rates for the same cortical regions were in the range of 6.24 Hz (percentiles, 75th = 10.37 Hz; 25th = 3.95 Hz). These differences were statistically significant when compared with K-Wallis and Bonferroni post hoc tests ($\chi^2 = 105.28$, $p < 0.001$).

Second, although it is normal to expect higher signal-to-noise ratios from sensory evoked responses under these conditions, the signals used to decode time were extracted from the population dynamics evoked over hundreds of milliseconds; that is, we used the information from the immediate sensory responses (~ 30 ms) but also from the following network responses that spanned for more than 200 ms after each stimuli (Fig. 1d). The following activity, characterized by sequential activation of cells and referred to as “packets of Information” (Luzsak et al, 2015), would be less affected by the potential signal-to-noise ratio confound.

Third, the possibility to decode time in our network state analysis depends mainly on two things. (1) The network representation on any stimulus within any given train must be different from the other stimuli of the same train. And (2), the network representation on any given stimulus, for example, stimulus #3, should be similar to stimuli #3 of the other trains of the experiment. This is reflected in the population trajectories depicted in the original Fig. 3c, especially for the striatum, where as time passes, each population trajectory (representing individual trials or trains) falls in distinguishable regions for every stimulus of the train (network states).

Finally, if the possibility to decode time could be masked (or enhanced) by differences in signal-to-noise ratios in the S1 or DLS, one would expect that removing from the analysis the moments where signal-to-noise ratio is higher, for example the immediate sensory responses to the stimuli, would significantly modify the accuracy to decode time. To test this possibility, we modified our original analysis. We artificially removed the spikes of each cell in the 50 milliseconds following each stimulus of every train of stimulation and repeated the decoding procedure in the modified data. The results are strikingly similar to our original analysis. Neither S1 nor DLS significantly improved or worsened their accuracy to decode time. This new analysis confirms two things: First, it is unlikely that the signal-to-noise ratio in our data is masking the ability to decode time from network activity in S1 (or striatum), and second, the accuracy to decode time depends mainly on the local network dynamics and not on the early sensory responses of the stimuli.

We have included the new analysis in Supplementary Fig. 4 e-g and added these arguments in the Results section (Page 6, line 32 to page 7 line 7).

2) The disconnection method, lesioning one area and other area contralaterally (Fig. 4b), is very complicated and tricky, although it was reported previously. In this method, the disconnection is defined as a symmetric abolishment of the same interareal connections on both sides. However, even though other

connections are also damaged on only one side, they are considered “functional” whether ipsi- or contralateral, ignoring any effect of such unilateral damages. For example, the VPL-DLS disconnection is actually accompanied by left S1-DLS disconnection and right VPL-S1 disconnection. Its assumption also ignores compensatory changes or side effects in the three and other (e.g., callosal) connections. In addition, the difference between the VPL-DLS disconnection and Partial disconnection (as a control) could be regarded as the presence/absence of an intact loop among the three areas on one side. Taken together, the VPL-DLS disconnection cannot be truly pathway-specific disconnection. This point should be clearly described in the manuscript.

We understand the perils of using the disconnection method. We are also well aware of the limitations in anatomical connections such as the one we are studying. For these reasons, we performed multiple “control groups” where the main possible forelimb sensory inputs to the DLS were targeted. These possibilities are schematized in Fig. 4a. Briefly, we followed the same rationale that the reviewer. Given the anatomical and electrophysiological evidence, sensory information from VPL could arrive through two pathways, directly from the VPL or through S1. Hence, lesioning the VPL on one side would not prevent information arriving from S1 from that same side (even when S1 would be expected to be diminished because of the VPL lesion). In the example of the reviewer, **“the VPL-DLS disconnection is actually accompanied by left S1-DLS disconnection and right VPL-S1 disconnection”**, for this very reason we performed the groups where S1 is disconnected from the DLS (orange code in Fig. 4) and the group where VPL is disconnected from S1 (blue code in Fig. 4). Nevertheless, we are perfectly aware of the limitation of the technique in this particular projection. We are not in a position of claiming that in our lesion experiments we achieve a “perfect pathway-specific” manipulation. We have modified the manuscript and explicitly stated the reviewer’s concerns in the discussion (Page 17, lines 3 to 8).

3) It is, therefore, necessary to show statistical comparisons of sensory spike responses of DLS neurons among these disconnection conditions in Sup. Fig. 2. More importantly, impaired sensory signals in the DLS neurons should be confirmed electrophysiologically during the task performance in those lesioned groups.

For the first part of the comment, we were able to perform electrophysiological confirmation of the lesions only after training was finished. Our recording technique in anesthetized animals under urethane prevented us from recovering those animals after the recording session. We have now included formal quantifications of the sensory representations in the cortex and striatum for those animals in Supplementary Fig. 2.

For the second part of the comment, **“impaired sensory signals in the DLS neurons should be confirmed electrophysiologically during the task performance in those lesioned groups”**.

We addressed this point in two ways. First, we were able to record a single naïve animal with a contralateral VPL-DLS lesion. We implanted a 64-channel silicon probe in the forelimb region of the primary somatosensory cortex, ipsilateral to the VPL lesion (probe centered in mm with respect to bregma: AP = 0.6, DL = 3.7). From this animal we were able to record 131 cells from 6 different depths between 0.9 to 1.7 mm during 6 sessions.

To further characterize the functionality of our manipulations during behavioral execution, we also propose a different method. We performed the same type of silicon probe high-density electrophysiological recordings in 3 freely moving expert animals executing the task. Aside from the silicon probe implantation, animals were infused in the VPL with the virus to express Channelrhodopsin 2 and implanted with two optic fibers directed to the VPL. We recorded 176 cells from the forelimb region of S1 (probe centered in mm with respect to bregma: AP = 0.6, DL = 3.7) from 9 different depths between 0.9 to 1.9 mm during 9 sessions. We also recorded 94 cells from the DLS from 12 different depths between 3.1 to 4.1 mm during 12 sessions. During recording sessions, 50% of randomly selected trials were optically stimulated. Behavioral effects during these trials were virtually identical to those in the animals reported in the original version of the MS (Supplementary Fig. 10 b-d).

To determine if a cell was modulated by the forelimb cycle movement, we calculated the correlation between the spiking activity of individual cells and the movement of the forelimb (contralateral to recording sites) during treadmill runs. We found that in general, the correlation values were significantly higher for cells recorded from non-lesioned animals than from lesioned animals (Supplementary Fig. 12e-g). Comparing spiking activity during stimulated vs. non-stimulated trials in the non-lesioned animals revealed that, in the cortex, 31% of the cells (27% in the striatum) significantly changed their firing rates with the optical stimulation of the VPL (Fig. 8 a-c). As depicted in raster plots and autocorrelograms in Supplementary Fig. 12 a and d, stimulation also produced changes in the patterns of activity. When specifically comparing the cells with higher spike-forelimb correlation values, we found that these values significantly decreased in both regions during stimulated trials. This effect was more evident in the DLS (Supplementary Fig. 12). We have included a description of these observations at the end of the Results (Page 13 line 23 to page 14 line 18) and Methods sections (Page 22 lines 13-15; page 23 lines 36-43).

Altogether, the anesthetized recordings after training (Supplementary Fig. 6) and the freely moving recordings in lesioned and non-lesioned animals with optogenetic manipulations (Supplementary Fig. 12) confirmed that we successfully modified the forelimb representations during the execution of the task. We hope that with the inclusion of the new analysis and experiments the reviewer found convincing evidence that our manipulations were satisfactory.

4) I would suggest simpler and more direct experiments using optogenetic inhibition to block the pathway signal specifically. Briefly, Halorhodopsin or Archaelhodopsin-T is expressed in the VPL neurons bilaterally by viral vector

infection, and their axonal terminals in the DLS are inactivated by local light illumination during task performance, which allows to inhibit the VPL-DLS pathway specifically. The optogenetic activation of VPL-DLS terminals with Channelrhodopsin-2 (Fig. 7e-h) will propagate back into other areas through axonal collaterals and can affect behavioral performance indirectly. Thus, it is not direct evidence of the pathway-specific function.

The reviewer is correct; in the behavioral experiments with Channelrhodopsin-2 stimulating terminals in the striatum, the pathway-specific interpretation may be compromised by the possibility of back-propagating action potentials. In the original version of the article we argued that given the short latencies of the action potentials evoked by this type of stimulation, it would be less probable that back-propagating action potentials would be contributing to responses recorded under anesthetized conditions. Nevertheless, we recognize we did not perform these experiments in freely moving animals, and hence the question remains open. To answer this concern, we performed the experiment proposed by Reviewer, i.e. we expressed Archærhodopsin in two over-trained animals (>5 months of training) and performed the same experiment as the one in Supplementary Fig.7, stimulating 50% of randomly selected trials in more than 5 sessions per animal. For the analysis we used > 300 stimulated and > 300 non-stimulated trials for each animal. The results are very similar to the ones reported for direct inhibition of the VPL (see the new Fig. 7). This confirms the pathway-specific function. Based on this comment and a previous comment from Reviewer 1, we have modified Fig. 7, originally dedicated only to Channelrhodopsin manipulations. Now we have summarized the results and show in the same figure the optogenetic inhibition and excitation experiments, since we believe it makes a better contrast for both manipulations.

5) The S1 looks more damaged than the DLS in the left MRI image of Sup. Fig. 4a. Sup. Fig. 4d shows certainly similar lesioned areas in the S1 and DLS. An additional analysis should be performed using only individuals with larger DLS lesion and less S1 lesion.

This analysis is partially included in the original version of the manuscript, where we calculated the correlations between performance and size of lesions. We observed that the size of DLS lesions statistically explained the variability of our data. Nevertheless, a simple correlation may not be the most appropriate solution because it will not consider interactions with the size of lesion in the cortex. Therefore, we performed multiple correlation analyses and confirmed that the variable that best explain the data are combinations of lesions VPL+DLS ($R = 0.89$; $p = 0.0107$); VPL+DLS+S1 ($R = 0.94$; $p = 0.0224$); DLS+S1 ($R = 0.66$; $p = 0.114$); VPL + S1 ($R = 0.19$; $p = 0.6518$).

6) Fig. 7 and Sup. Fig. 7. Did the repetitive stimulation at 3.3 or 5 Hz indeed affect the frequency (Hz) of locomotion cycles as a temporal reference? If it improved the task score (entry time) without changing the locomotion cycles, what does the repetitive sensory responses mean in Sup. Fig. 2?

Stimulation did not disturb locomotion cycles or speed of movement; the most plausible explanation is that we stimulated a sensory rely. On the other hand, the motor component is not present during anesthetized experiments; our stimulations did not produce active movements.

Minor comments:

1) Fig. 1 legend (p.23). Panel letters “b” to “h” for Fig. 1 panels were shifted in the legend.

Corrected.

2) Fig. 1a and Sup. Fig. 1b-d. FG-positive should be unified to either lighter (white, fluorescence) or darker (black, reversed).

Changed as suggested.

3) Fig. 2a, Fig. 7a,e, and Sup. Fig. 7a,f. Font size is too small. Is each photo a combination of two photos on the right and left sides? If so, they should be separated with a gap between them or by labeling. Also, the direction should be indicated by R and L or M and L on these photos.

Changed as suggested.

4) Fig. 2d. Gray for mechanic is too faint to see. It should be darker.

Changed as suggested.

5) Fig. 3a-c. I cannot see “stim. 5” too, which should be darker.

Changed as suggested.

6) “VPL-DLS disconnection”, etc. are misleading as I mentioned above. I would suggest more correct wording, for example, “cVPL & DLS lesion” (c, contralateral).

Changed as suggested.

7) Sup. Fig. 4a,b,c. The lesioned areas should be marked by arrows or arrowheads.

Changed as suggested.

8) Sup. Fig. 2. I would prefer to move Sup. Fig. 2 behind Sup. Fig. 4.

Changed as suggested.

Reviewers' Comments:

Reviewer #1:

Remarks to the Author:

The authors have carefully addressed all my concerns. Consequently this revision is much improved and acceptable for publication.

Reviewer #2:

Remarks to the Author:

The authors have addressed my original comments adequately.

I have two new comments:

1) The revised manuscript mentions the possibility of a VLP-to-Pf projection as a potential explanation for histologic observations. The idea of intra-thalamic projections between nuclei was a major topic of debate for decades in the 1900's. The conclusion of that debate was widespread acceptance that intra-thalamic connections between nuclei do not exist. Jones does a nice job of laying out the evidence and history in his book on the thalamus. The point here being that the authors invoke an idea that was long debated and then laid to rest. Resurrection of that idea should be supported by strong evidence.

2) The results from sensory stimulation in anesthetized animals (Fig. 3) are thought provoking. Presumably the observed representation of stimulus order reflects inherent properties of striatal activity. E.g., these are naive unconscious animals in whom learning is unlikely to have occurred. (Nevertheless, it would be helpful to know if the pattern of responses of individual neurons across the 5 stimuli in the train evolves across repetitions of the train.) If true, this raises questions about the types of stimulus trains that striatum can represent in this fashion. What would striatum do with long stimulus trains (e.g., a series of 10 or more stimuli rather than 5)? or if the time interval between successive trains was shorter or longer than the stated 5-10 sec? It is reasonable to predict that the striatum's ability to encode progression across a stimulus trains is limited to a certain range of train parameters (e.g., intra-train length and frequency and inter-train interval).

Those questions are beyond the scope of the current manuscript, but it does not seem too much to ask that the manuscript offer a proviso that the anesthetized and awake behaving approaches are not necessarily tapping in to identical physiologic processes. The results from anesthetized prep. presumably reflect inherent dynamics of a naive anesthetized neural network that includes the striatum. In contract, the processes being studied in behaving and learning animals are quite different.

Minor comments: The manuscript contains multiple grammatical and typographic errors. Most of the ones I found are in the revisions marked in red in the submitted manuscript. Presumably these problems will be caught by the copy editing staff at the journal.

Reviewer #3:

Remarks to the Author:

This work has been greatly improved by performing additional experiments using optogenetic inhibition with Archaelhodopsin and some additional analyses. Their descriptions have also been modified carefully and correctly all through the manuscript. In my comment 3, what I meant was that they should statistically compare the sensory spike response of DLS neurons among the different

disconnection conditions in unit-recording experiments similar to Suppl. Fig. 2. Training or anesthesia does not matter. After all, the additional optogenetic data have directly solved the problem. Now, I am convinced of the novelty and importance of this work. Thank you.

Reviewer #1 (Remarks to the Author):

The authors have carefully addressed all my concerns. Consequently, this revision is much improved and acceptable for publication.

We thank the reviewer for the time and dedication and her/his thoughtful comments, addressing them significantly improved the presentation of our work.

Reviewer #2 (Remarks to the Author):

The authors have addressed my original comments adequately.

I have two new comments:

1) The revised manuscript mentions the possibility of a VLP-to-Pf projection as a potential explanation for histologic observations. The idea of intra-thalamic projections between nuclei was a major topic of debate for decades in the 1900's. The conclusion of that debate was widespread acceptance that intra-thalamic connections between nuclei do not exist. Jones does a nice job of laying out the evidence and history in his book on the thalamus. The point here being that the authors invoke an idea that was long debated and then laid to rest. Resurrection of that idea should be supported by strong evidence.

We thank the reviewer for this comment. We agree, we originally included this statement in response to a previous comment from another reviewer, but we understand that such an asseveration must be accompanied by stronger experimental evidence. In response to this comment we have modified this statement focusing only in the main message, that is, our infections did not contaminate the Pf, but without suggesting a VPL-Pf projection. Page 4, Lines 13-14.

2) The results from sensory stimulation in anesthetized animals (Fig. 3) are thought provoking. Presumably the observed representation of stimulus order reflects inherent properties of striatal activity. E.g., these are naive unconscious animals in whom learning is unlikely to have occurred. (Nevertheless, it would be helpful to know if the pattern of responses of individual neurons across the 5 stimuli in the train evolves across repetitions of the train.) If true, this raises questions about the types of stimulus trains that striatum can represent in this fashion. What would striatum do with long stimulus trains (e.g., a series of 10 or more stimuli rather than 5)? or if the time interval between successive trains was shorter or longer than the stated 5-10 sec? It is reasonable to predict that the striatum's ability to encode progression across a stimulus trains is limited to a certain range of train parameters (e.g., intra-train length and frequency and inter-train interval).

Those questions are beyond the scope of the current manuscript, but it does not seem too much to ask that the manuscript offer a proviso that the anesthetized and awake behaving approaches are not necessarily tapping in to identical physiologic processes. The results from anesthetized prep. presumably reflect

inherent dynamics of a naive anesthetized neural network that includes the striatum. In contrast, the processes being studied in behaving and learning animals are quite different.

The questions from the first part of the comment are certainly interesting; the anesthetized preparation is indeed a useful tool to study the inherent properties of striatal networks. It's also true that conclusions extracted from this preparation cannot be directly exported to freely behaving animals. For this reason, we conducted a large body of experiments that lead to the conclusions of this manuscript.

On the other hand, regarding the experiments suggested by the reviewer, we completely agree. In fact, to further explore the inherent properties of this network, as well as the contribution of motor and sensory inputs from cortex and thalamus, we are already working on experiments similar to the ones proposed. We hope to have clear answers in following publications.

The last part of the reviewer's comment, on the anesthetized *versus* awake striatal neural dynamics, is very similar to a previous comment from Reviewer #1, and we had already included a specific paragraph in the discussion of the revised version of the MS to address this issue, there we wrote:

"An important question from our results is whether under these anesthetized conditions, sensory inputs to the DLS may sustain temporal representations in longer timescales, like those reported for the range of dozens of seconds to minutes^{26,43}. This possibility would need to be addressed experimentally but it is likely that those longer temporal representations are based on a more complex mechanism, since they were reported in conditions where animals were not necessarily implicated in a particular sequence of movements or a stereotyped behavior that would produce a stable, rhythmic sensory flow. In this regard, our data are more compatible with the idea that sensory-based temporal representations are fundamental for the learning and execution of short sequences of movement in the range of seconds"

Minor comments: The manuscript contains multiple grammatical and typographic errors. Most of the ones I found are in the revisions marked in red in the submitted manuscript. Presumably these problems will be caught by the copy editing staff at the journal.

We have revised MS and corrected these mistakes.

Reviewer #3 (Remarks to the Author):

This work has been greatly improved by performing additional experiments using optogenetic inhibition with Archaelhodopsin and some additional analyses. Their descriptions have also been modified carefully and correctly all through the manuscript. In my comment 3, what I meant was that they should statistically compare the sensory spike response of DLS neurons among the different disconnection conditions in unit-recording experiments similar to Suppl. Fig. 2. Training or anesthesia does not matter. After all, the additional optogenetic data have directly solved the problem. Now, I am convinced of the novelty and importance of this work. Thank you.

We thank the reviewer for his comments, her/his suggestions significantly improved our work.